# Utility of Deep Learning Algorithms in Initial Flowering Period Prediction Models

Guanjie Jiao [1], Xiawei Shentu [2], Xiaochen Zhu [2,*], Wenbo Song [3], Yujia Song [2] and Kexuan Yang [2]

1 School of Environmental Science and Engineering, Nanjing University of Information Science and Technology, Nanjing 210044, China
2 School of Applied Meteorology, Nanjing University of Information Science and Technology, Nanjing 210044, China
3 School of Artificial Intelligence (School of Future Technology), Nanjing University of Information Science and Technology, Nanjing 210044, China
* Correspondence: xiaochen.zhu@nuist.edu.cn

**Abstract:** The application of a deep learning algorithm (DL) can more accurately predict the initial flowering period of *Platycladus orientalis* (L.) Franco. In this research, we applied DL to establish a nationwide long-term prediction model of the initial flowering period of *P. orientalis* and analyzed the contribution rate of meteorological factors via Shapely Additive Explanation (SHAP). Based on the daily meteorological data of major meteorological stations in China from 1963–2015 and the observation of initial flowering data from 23 phenological stations, we established prediction models by using recurrent neural network (RNN), long short-term memory (LSTM) and gated recurrent unit (GRU). The mean absolute error (MAE), mean absolute percentage error (MAPE), and coefficient of determination ($R^2$) were used as training effect indicators to evaluate the prediction accuracy. The simulation results show that the three models are applicable to the prediction of the initial flowering of *P. orientalis* nationwide in China, with the average accuracy of the GRU being the highest, followed by LSTM and the RNN, which is significantly higher than the prediction accuracy of the regression model based on accumulated air temperature. In the interpretability analysis, the factor contribution rates of the three models are similar, the 46 temperature type factors have the highest contribution rate with 58.6% of temperature factors' contribution rate being higher than 0 and average contribution rate being $5.48 \times 10^{-4}$, and the stability of the contribution rate of the factors related to the daily minimum temperature factor has obvious fluctuations with an average standard deviation of $8.57 \times 10^{-3}$, which might be related to the plants being sensitive to low temperature stress. The GRU model can accurately predict the change rule of the initial flowering, with an average accuracy greater than 98%, and the simulation effect is the best, indicating that the potential application of the GRU model is the prediction of initial flowering.

**Keywords:** *P. orientalis*; recurrent neural network; inverse distance weighting; accumulated air temperature

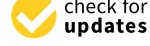



## 1. Introduction

Flowering is one of the sensitive indicators for assessing climate change [1–6], which reflects changes in surface vegetation and eco-health [7]. Moreover, flowering has tremendous economic value; plants with short flowering time displays have promoted the development of tourism, and tourism activities characterized by flower viewing have gradually become important cultural events, and the market is constantly expanding [8–10].

The climatic conditions have an impact on the initial flowering period of plants, and air and soil temperature are the main factors [5,11–13]. Important progress has been made in phenological research on flowering forecasting based on meteorological data, which has mainly established statistical equations to predict flowering period based on the correlation between meteorological data and phenological data [3,14,15]. In 1974, Richardson et al. [16]

first proposed the application of the chill unit model to research on peach tree dormancy prediction, which calculates the chill unit accumulation coinciding with the completion of plant dormancy to evaluate the impact of low temperature in winter on flowering and to predict the initial flowering period. In 1979, White [17] constructed a linear regression model based on phenological data from 53 species of Montana, which support subsequent flowering studies. In 1986, Anderson et al. [18] further obtained the ASYMCUR GDH model that is an improved normal plant model to fit growing divide hour (GDH) responding to the environment on the basis of the chill unit model, and carried out research on the prediction of the tart cherry flowering period. In 1998, to avoid damage to plants caused by frost and hazards brought by climate change, such as rising temperature, Hakkinen et al. [19] used nearly 60 years of phenological data of birth bud observation in southern Finland from 1896–1955 and meteorological data of light signal and air temperature to predict the bud burst of birch trees by the light and temperature driven model. In recent years, as data work has continued to improve, flowering forecasting has begun to focus on accurate predictive models applied to a wide range of flowers. In 2004, Demeloabreu [20] carried out flowering prediction for different olive varieties in multiple regions to analyze the impact of global warming on olive production. Soil moisture is an important factor affecting spring phenology, so Yashvir et al. [21] utilized soil moisture as a correction factor to improve the accuracy of the original chickpea flowering prediction model in 2019. The research on flowering period in China focuses on analyzing the mechanism of meteorological influence on flowering. In 2019, Wu D et al. [22] conducted analysis through the forecasting model of apple flowering in Shaanxi, which refers to prediction of flowering period at different stations and analysis of the applicability by using the mechanistic models to simulate the growth process of phenology. In 2021, Tan J et al. [23] conducted a fine fitting analysis of cherry flowering and concluded that air temperature and precipitation are the main impact factors of previous cherry period research at Wuhan University.

Current research on flowering forecasting has problems, such as the limited time and space range of accurate predictions and uncertainty around meteorological factors affecting flowering, and currently the demand for initial flowering periods of plants in the Chinese flowering market covers the whole country. A solution for spatial phenology modelling may be to model phenology using herbarium and Citizen Science records and gridded climatic data. Recently, the flowering of Anemone nemorosa was modelled in this way across Europe. However, this approach has some limitations related to the availability of replicated phenological observations and spatial and taxonomic biases [24]. Hence, long-term local monitoring data are still invaluable in phenological studies. With the in-depth integration of artificial intelligence (AI) and meteorological big data, more scholars have begun to pay attention to the application of machine learning (ML) in phenology [25], but detailed research on deep learning (DL) in flowering prediction is lacking.

In our research, we demonstrate the capabilities of deep learning algorithms that have so far been used to a limited extent in phenological research. We believe that the results obtained in our study will find wide application and contribute to a better understanding of the phenological response of plants to meteorological conditions. We also analyze the contribution of each factor via Shapely Additive Explanation (SHAP) to interpret the deep learning model. We expect to provide a scientific basis for nationwide long-term, data-driven flowering prediction models based on our research.

## 2. Materials and Methods

### 2.1. Studied Species

*P. orientalis* (Cupressaceae) is also named tujia or arborvitae. Its initial flowering period is from March to April, and its cones mature in October.

*P. orientalis* has good stress resistance, which can withstand various extreme environmental conditions [26,27], such as drought, high temperature and low temperature stress, etc. However, the geographical advantages of abundant rainfall and high humidity in southern China can ensure its more healthy growth [28].

*P. orientalis* is one of the most widely distributed plants produced in southern Inner Mongolia, Jilin, Liaoning, Hebei, Shanxi, Shandong, Jiangsu, Zhejiang, Fujian, Anhui, Jiangxi, Henan, Shaanxi, Gansu, Sichuan, Yunnan, Guizhou, Hubei, Hunan, northern Guangdong and northern Guangxi in China [29].

*2.2. Region*

The research is to establish prediction models of initial flowering period of *P. orientalis* in China in the Chinese region (73°33′ E-135°05′ E, 3°51′ N-53°33′ N). Due to the large area, the distribution of meteorological elements in China is complex, mainly reflected in the uneven distribution of air temperature, precipitation and humidity, etc.

Temperature regions are divided by accumulated temperature. In China, there are five temperature regions of tropical, subtropical, warm temperate, mesotemperate and cold, whose accumulated temperature value is increasing from north to south, so lower latitude affects the growth process of plants less. There is also a special Qinghai Tibet Plateau region influenced by high altitude of an average 4000 m [30].

According to the humidity index (HI), China can be divided into four regions, namely, arid region (AR), semi-arid region (SAR), semi-humid region (SHR) and humid region (HR) [31]. And HI can reflect the regional humidity, which affects the physiological process of plants through the influence on the water potential, which is the key to the process of plant water absorption. The partition result can be obtained in Figure 1.

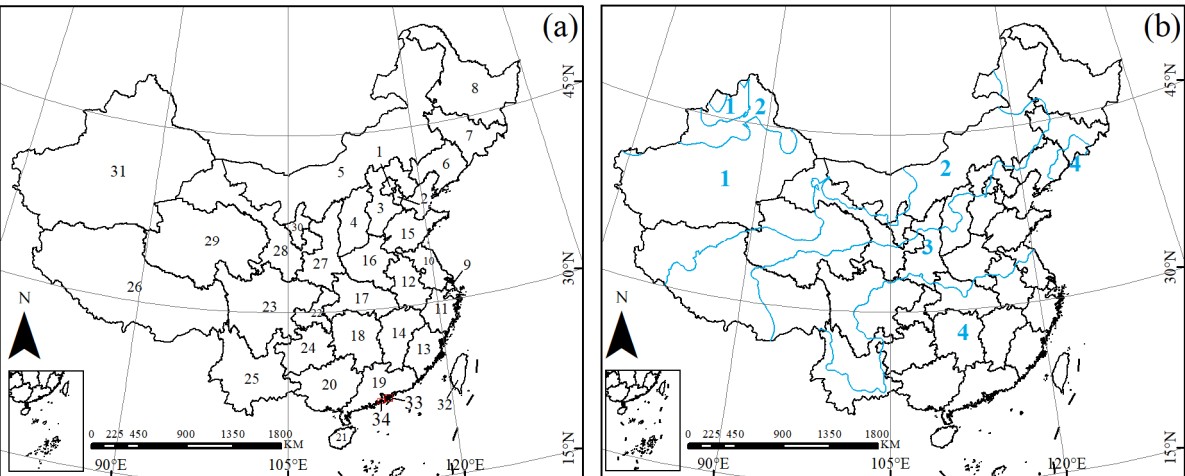

**Figure 1.** China's regional map of (**a**) administrative division, and (**b**) arid-humid division. Beijing, Tianjing, Hebei, Shanxi, Inner Mongolia, Liaoning, Jilin, Heilongjiang, Shanghai, Jiangsu, Zhejiang, Anhui, Fujian, Jiangxi, Shandong, Henan, Hubei, Hunan, Guandong, Guangxi, Hainan, Chongqing, Sichuan, Guizhou, Yunnan, Tibet, Shaanxi, Gansu, Qinghai, Ningxia, Xinjiang, Taiwan, Hongkong and Macao are denoted by numbers from 1 to 34 in (**a**), respectively. The arid region (AR), semi-arid region (SAR), semi-humid region (SHR) and humid region (HR) are denoted by numbers 1, 2, 3 and 4, respectively.

The analysis of the impact of China's meteorological element conditions on the spatial distribution of *P. orientalis* in the initial flowering period is regional, so we introduced China's administrative division to help spatial analysis. The vector diagram of the division of administrative regions in China is derived from the National Platform for Common Geospatial Information Services (https://www.tianditu.gov.cn (accessed on 8 September 2022)).

*2.3. Materials*

Phenological data are observational data that reflect periodic biological phenomena including initial flowering period, which refers to the time of one or few flowers fully open. To obtain enough data for DL training, we collected the initial flowering data of *P. orientalis*

from the National Earth System Science Data Center (https://geodata.cn/ (accessed on 13 July 2022)) and the Earth Big Data Science Data Center of the Chinese Academy of Sciences (https://data.casearth.cn/ (accessed on 11 August 2022)) and selected available data that included city stations in Baoding, Beijing, Changde, Guiyang, Hohhot, Shanghai, Foshan, Nanjing, Nanchang, Hefei, Harbin, Kunming, Guilin, Wuhan, Minqin, Shenyang, Tai'an, Xi'an, Chongqing, Yinchuan, Changchun, Changsha and Yancheng from 1961–2015. The spatial distribution can be obtained in Figure 2. A total of 357 valid data points were obtained.

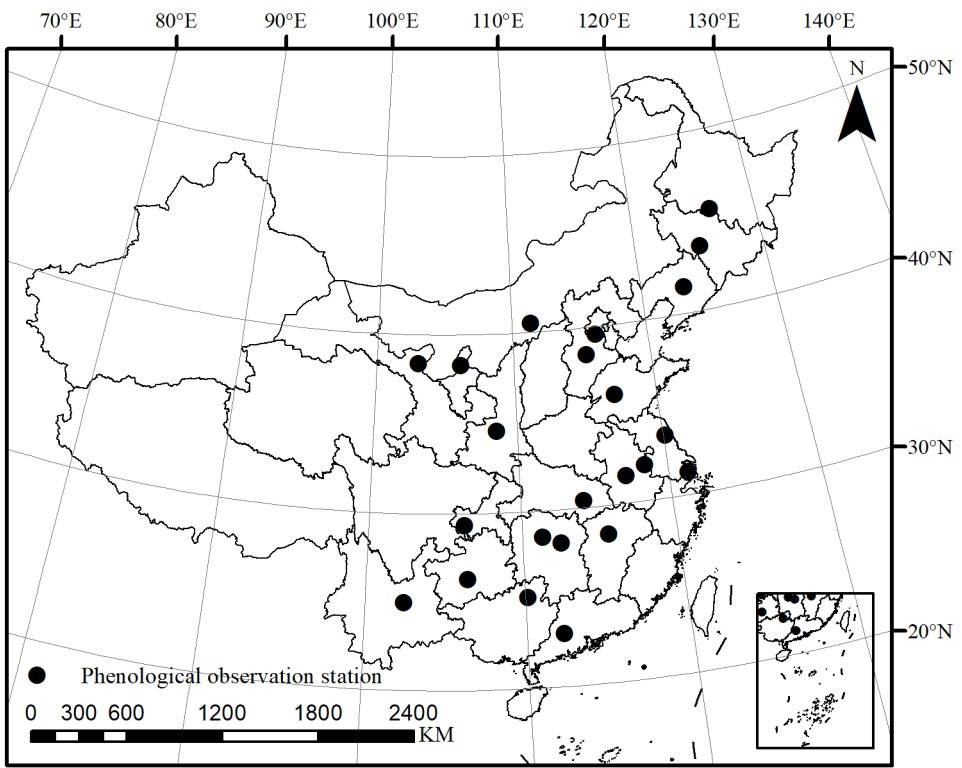

**Figure 2.** Geographical distribution map of phenological stations.

The meteorological data were obtained from the China Meteorological Science Data Sharing Network "China Ground Meteorological Data Dataset V3.0". A total of 23 basic city stations were selected, and we obtained the meteorological elements of average temperature (°C), daily minimum temperature (°C), daily maximum temperature (°C), daily average ground temperature (°C), daily average precipitation (mm), daily average sunshine hours (h), daily average relative humidity (%), and daily average pressure (Pa) from 1 January to 30 April 1961 to 2015 in these stations.

*2.4. Methods*

2.4.1. Selection of Meteorological Factors

The effect of air temperature on the initial flowering period is most pronounced, followed by sunshine and precipitation [11,21]. In ecological research, crop growth and development need to accumulate to a certain sum of temperature, so the air temperature is usually expressed in cumulative amount, which is referred to as the accumulated temperature. According to different time scales, the action time of accumulated temperature is varied. In the process of growth, crops respond to the temperature limit, which is the lower limit temperature. When the temperature is lower than the lower limit temperature, the plants will not grow and develop. The accumulated amount of temperature above the lower limit temperature is the active accumulated temperature, and the accumulated difference between the temperature and the lower limit temperature is the effective accumulated temperature, which can be applied to air temperature and ground temperature.

$$Effective\ accumulated\ temperature = \sum (T_i - C_0) \tag{1}$$

$$Accumulated\ temperature = \sum T_i \tag{2}$$

$$Active\ accumulated\ temperature = \sum T_i \quad T_i \geq C_0 \tag{3}$$

where $T_i$ is the daily average temperature, and $C_0$ is the lower limit temperature.

Since the initial flowering period of Platycladus orientails is mainly in the middle of April, we focused on the meteorological data from January to April. During data processing, we read the meteorological data from each station and used 0 °C, 3 °C, 5 °C AND 10 °C as the lower limit temperatures to calculate the effective accumulated temperature and counted the accumulated temperature and average temperature from January to early April for ten days and the average ground temperature monthly and other factors, as detailed in Table 1.

**Table 1.** Table of meteorological factors affecting the initial flowering of *P. orientalis*.

| Meteorological Elements | | Meteorological Factors | Number of Factors |
|---|---|---|---|
| Temperature | 1. | The effective cumulative temperature of 0 °C, 3 °C, 5 °C, 10 °C (°C); | 46 |
| | 2. | Active temperature (°C); | |
| | 3. | Accumulated temperature (°C); | |
| | 4. | Accumulated temperature for ten days (°C); | |
| | 5. | Average temperature for ten days (°C); | |
| | 6. | Days when the minimum/maximum temperature is less than 0 °C, 5 °C, 10 °C (d); | |
| | 7. | Days when the minimum/maximum temperature is more than 0 °C, 5 °C, 40 °C (d); | |
| | 8. | Average monthly minimum/maximum temperature from January to April (°C). | |
| Ground temperature | 1. | Accumulate ground temperature (°C); | 7 |
| | 2. | Average monthly ground temperature from January to April (°C); | |
| | 3. | Days when the ground temperature is less than 0 °C (d); | |
| | 4. | Days when the ground temperature is more than 40 °C (d). | |
| Precipitation | 1. | Cumulative precipitation (mm); | 10 |
| | 2. | Average precipitation (mm); | |
| | 3. | Accumulated monthly precipitation from January to April (mm); | |
| | 4. | Average monthly precipitation from January to April (mm). | |
| Hours of sunshine | 1. | Total hours of sunshine (h); | 5 |
| | 2. | Monthly hours of sunshine from January to April (h). | |
| Relative humidity | 1. | Average relative humidity (%); | 11 |
| | 2. | Average relative humidity for ten days (%). | |
| Pressure | 1. | Average pressure for ten days (hPa). | 10 |

Because different meteorological data have different degrees of influence [23], we considered different time resolutions when establishing meteorological factors. For example, we mainly deal with accumulated temperature for ten days when doing accumulated temperature calculation through Equation (2), which is a method to calculate ten days of the month. Therefore, each month will have three different accumulated temperatures for ten days values, which are divided into an early value, middle value and late value.

### 2.4.2. Data Processing

For the convenience of comparison between two different years, we use the data of ordinal number from 1 January to the current date as phenological data of flowering.

With each meteorological factor as the independent variable and ordinal number as the dependent variable, a phenological-meteorological dataset is constructed, and the dataset is normalized to facilitate weight distribution in the deep learning model. At a ratio of 7:3, we divided the training dataset and test dataset for model training and modelled effect evaluation to ensure sufficient samples during training, whose distribution is the same and not repeated, to evaluate the quality of model training.

In order to make each factor value dimensionless in the process of DL training, we normalized the data by max–min method, which will limit each data point to 0–1.

$$y' = \frac{y - \min}{\max - \min} \tag{4}$$

where $y'$ is normalized value, $y$ is value to be normalized, min is the minimum value of the same value and max is the maximum value of the same value.

### 2.4.3. Deep Learning Model

In current prediction research, such as Southern Oscillation, local evaporation and drought prediction, the deep learning algorithm has a better fitting ability and can improve the spatial resolution of prediction [32,33]. Compared with other common networks such as convolutional neural network (CNN) and artificial neural network (ANN), recurrent neural networks have a significant role in time series processing. The initial flowering period is predicted by three common deep learning prediction models, namely, the recurrent neural network (RNN), long short-term memory (LSTM), and the gated recurrent unit (GRU).

- Compared with other neural networks, the RNN can predict the current input value by combining the input values of the first N time series, that is, it has correlation in the time series.
- LSTM can learn the long-term dependence between two variables and retain the error, which can be maintained at a constant level when backpropagation is carried out along the time layer [34,35]. LSTM is equipped with three gating devices to filter the input data, namely, the input gate, forget gate and output gate. The forget gate will generate a value between 0 and 1 according to the output and current input of the previous time to decide whether to retain the information of the previous time [35]. The time function of the forget gate is mainly controlled by the sigmoid activation function:

$$f_t = \sigma(W_f \cdot [h_t - 1, x_\mathrm{t}] + b_f) \tag{5}$$

where $f$ is the forget gate, $W$ is the weight matrix, $b_f$ is the offset term, and $\sigma$ is the sigmoid activation function. The closer the value of $f_t$ is to 0, the more items will be forgotten.

- Compared with the LSTM model, the GRU simplifies the calculation steps and substantially increases the training speed, while the GRU also uses a gate device to filter information, namely, the reset gate and update gate. In the process of training, the input information will not be cleared by the gate device, but the necessary information will be retained in the next cycle, and the information will be saved to avoid the problem of gradient disappearance. Since there are only two gate structures, the actual running time of the GRU model is substantially less than that of LSTM with fewer network parameters, so the risk of GRU model overfitting is smaller under the condition of ensuring accuracy.

### 2.4.4. Training Effect Indicators

The mean squared error (MSE) is used as a loss function, and the mean absolute error (MAE), mean absolute percentage error (MAPE) and coefficient of determination ($R^2$) are utilized as the training effect indicators to evaluate the model performance.

$$\text{MSE} = \frac{1}{m} \sum_{i=1}^{m} (y_i - \hat{y}_i)^2 \tag{6}$$

$$\text{MAE} = \frac{1}{m} \sum_{i=1}^{m} |(y_i - \hat{y}_i)| \tag{7}$$

$$\text{MAPE} = \frac{100\%}{m} \times \sum_{i=1}^{m} \left| \frac{\hat{y}_i - y_i}{y_i} \right| \tag{8}$$

$$R^2 = 1 - \frac{\sum_{i=1}^{m} (y_i - \hat{y}_i)^2}{\sum_{i=1}^{m} (y_i - \overline{y}_i)^2} \tag{9}$$

where $y_i$ is the true value, $\hat{y}_i$ is the predicted value, $m$ is the number of samples, and $\overline{y}_i$ is the mean of the prediction.

MSE has high robustness, and it can effectively converge with a fixed learning rate, so the model with MSE as the loss function can maintain the accuracy in the process of convergence compared with the model with MAE as loss function [36]. MAE and MAPE are commonly employed indicators to reflect the degree of deviation between the predicted value and the true value. $R^2$ is mainly used to judge the linear relationship between the model prediction and the true value. Therefore, when the value of $R^2$ is near 1, the simulation degree of the model is accurate. The above four indicators are applied as mathematical definitions in general statistical research, so they are highly recognized.

### 2.4.5. Interpretability Model Based on SHAP

Shapely Additive Explanation (SHAP) is a method which uses game theory that is used to study the mathematical theory of contribution rate as the ideological carrier to calculate the impact of the characteristic variables of sample data on the results of the prediction model and then to measure the contribution of these characteristic variables. This approach explains the CART-based complex integrated learning model [37].

The core of SHAP is to calculate the Shapley value of variables, which represents the importance of determining the influence of various factors on the prediction.

$$\phi_i = \sum_{S \subseteq M \setminus \{i\}} \frac{|S|!(|M| - |S| - 1)!}{|M|!} [f_x(S \cup \{i\}) - f_x(S)] \tag{10}$$

where $M$ denotes all feature sets $S$ represents subsets of $i$, $f_x(S \cup \{i\})$ is the predicted value of the characteristic variable containing only $S \cup \{i\}$ in the sample data, and has a Shapley value of $i$.

As the complexity of using the Shapley value to traverse all subsets exponentially increases, this leads to an excessively long computing time and increases the computational burden, Lundberg and Lee proposed the Tree SHAP model based on the tree model in machine learning combined with the Shapley value [37]. In this research, we used the Deep SHAP model interpreter to rank the contribution of 89 meteorological factors that affect the initial flowering of *P. orientalis*. The Deep SHAP model avoids heuristic selection of linearized components but enables effective linearization from the SHAP values calculated for each component [37]. Therefore, the contribution of different factors in each sample to the model prediction can be achieved via Deep SHAP.

2.4.6. Overall Process of Predicting the Initial Flowering Period in DL

Based on the phenological observation city network and the meteorological observation data of China, we built a comprehensive dataset of the initial flowering and meteorology, importing the dataset into the RNN, LSTM, and GRU models as input vectors and using MSE as the loss function. When the loss function converges, the model is considered mature. MAE, MAPE and $R^2$ are selected as evaluation indicators to express the prediction effect. In order to compare the difference between the initial flowering period prediction model based on deep learning algorithms and the traditional flowering prediction models, we selected the multiple linear regression model based on accumulated air temperature as the representative of the traditional initial flowering period prediction model, and compared the prediction effect of this model with DL The interpretability model based on SHAP is adopted to further analyze the interpretability and stability of the model. This process can be obtained in Figure 3.

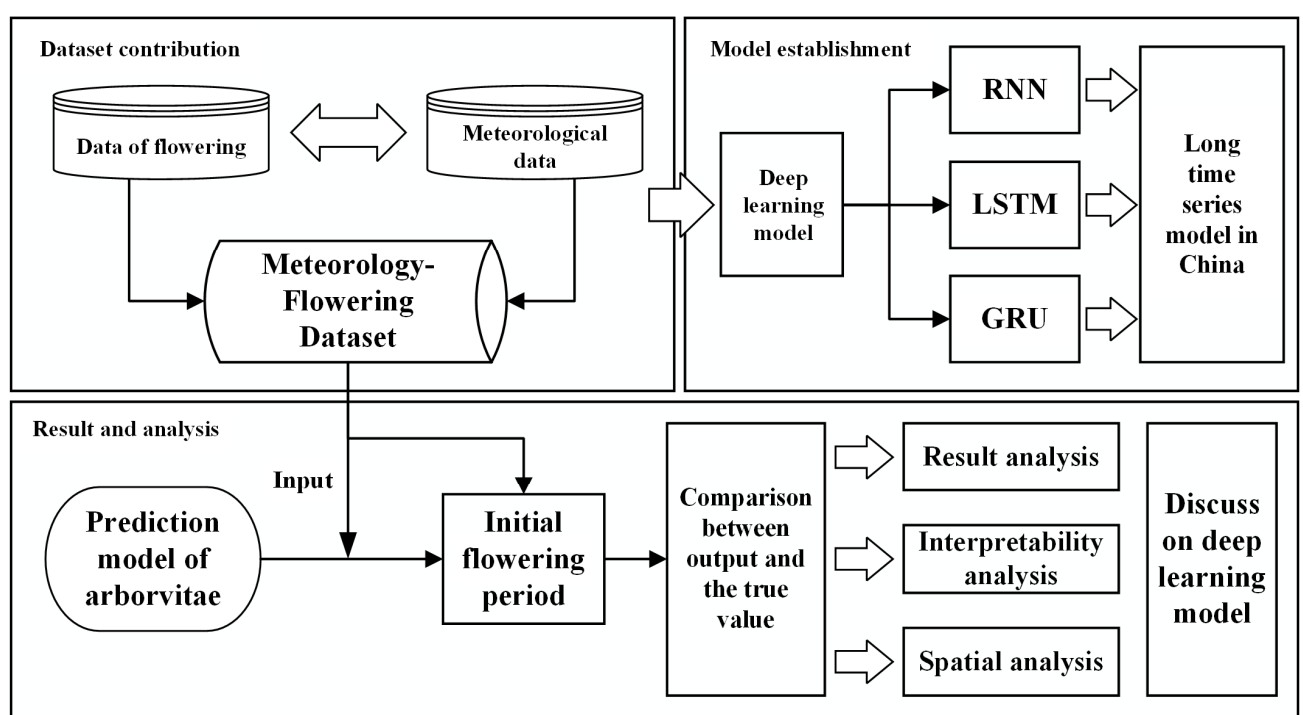

**Figure 3.** Flow figure of establishing DL models.

### 3. Results

*3.1. Basic Characteristics of P. orientalis during Initial Flowering*

As shown in Table 2, the flowering period of *P. orientalis* has obvious regional characteristics: with an increase in latitude, the average initial flowering period is gradually postponed, and the ordinal number of cities in northeast China is nearly 80 d (as a unit representing days), higher than that of coastal cities in south China such as Foshan and Shanghai etc., which is related to the generally high light, temperature and precipitation resources in south China. The dispersion degree of the initial flowering period of different stations can be obtained from the standard deviation. The standard deviation of 23 stations is concentrated at approximately 10 d. The maximum of Kunming station is 23.93 d, and the minimum of Harbin station is 1.50 d. Among all the data, the ordinal number of the earliest flowering period is 5 d observed at Kunming station, and that of the latest flowering period is 136 d at Minqin station. There are obvious interannual fluctuations and spatial differences in the observation data of each station, and the degree of dispersion is large with the range is 131 and normalized standard deviation is more than 0.2. Therefore, it is necessary to establish an accurate prediction model to effectively predict the initial flowering of *P. orientalis* nationwide.

**Table 2.** Table of ordinal number information of *P. orientalis*' initial flowering period.

| Station | Average Value (d) | Minimum Value (d) | Maximum Value (d) | Range (d) | Standard Deviation (d) | Skewness | Kurtosis |
|---|---|---|---|---|---|---|---|
| Baoding | 95.00 | 76 | 111 | 35 | 10.29 | −0.16 | −0.28 |
| Beijing | 86.97 | 65 | 108 | 43 | 10.03 | 0.12 | −0.59 |
| Changde | 59.88 | 38 | 78 | 40 | 10.06 | −0.27 | 0.026 |
| Guiyang | 57.05 | 33 | 86 | 53 | 13.89 | −0.11 | −0.44 |
| Hohhot | 108.00 | 101 | 121 | 20 | 6.31 | 0.97 | 0.19 |
| Shanghai | 63.38 | 50 | 76 | 26 | 7.61 | −0.04 | −0.07 |
| Foshan | 48.78 | 32 | 65 | 33 | 12.27 | −0.08 | −1.88 |
| Nanjing | 44.90 | 31 | 55 | 24 | 7.30 | −0.36 | −0.69 |
| Nanchang | 55.78 | 25 | 76 | 51 | 13.43 | −0.58 | 0.34 |
| Hefei | 63.93 | 41 | 78 | 37 | 11.11 | −0.67 | −0.70 |
| Harbin | 130.50 | 129 | 132 | 3 | 1.50 | 0.01 | 0.01 |
| Kunming | 40.08 | 5 | 98 | 93 | 23.96 | 0.93 | 0.95 |
| Guilin | 43.35 | 22 | 74 | 52 | 16.51 | 0.80 | −0.33 |
| Wuhan | 88.05 | 52 | 112 | 60 | 18.94 | −0.41 | −1.17 |
| Minqin | 104.93 | 92 | 136 | 44 | 10.76 | 1.61 | 4.07 |
| Shenyang | 111.20 | 104 | 122 | 18 | 7.33 | 0.69 | −2.49 |
| Tai'an | 76.25 | 70 | 86 | 16 | 6.01 | 1.29 | 1.78 |
| Xi'an | 65.86 | 46 | 81 | 35 | 8.20 | −0.43 | 0.51 |
| Chongqing | 54.62 | 24 | 76 | 52 | 14.99 | −0.39 | −1.05 |
| Yinchuan | 110.21 | 84 | 123 | 39 | 12.90 | −0.83 | −0.67 |
| Changchun | 111.96 | 93 | 129 | 36 | 7.91 | 0.12 | 0.89 |
| Changsha | 54.00 | 45 | 63 | 18 | 9.00 | 0.01 | 0.01 |
| Yancheng | 68.09 | 44 | 80 | 36 | 8.09 | −1.09 | 1.69 |

### 3.2. Model Training Effect

Normalized meteorological data and initial flowering data were imported into the DL models as inputs. It can be seen from Figure 4 that with the increase in the training epoch which represents the number of cycles in the training process, the loss functions of the three deep learning models converge, which shows that the prediction error of the model reaches a small value. Therefore, the training is stopped, and the flowering prediction test is conducted.

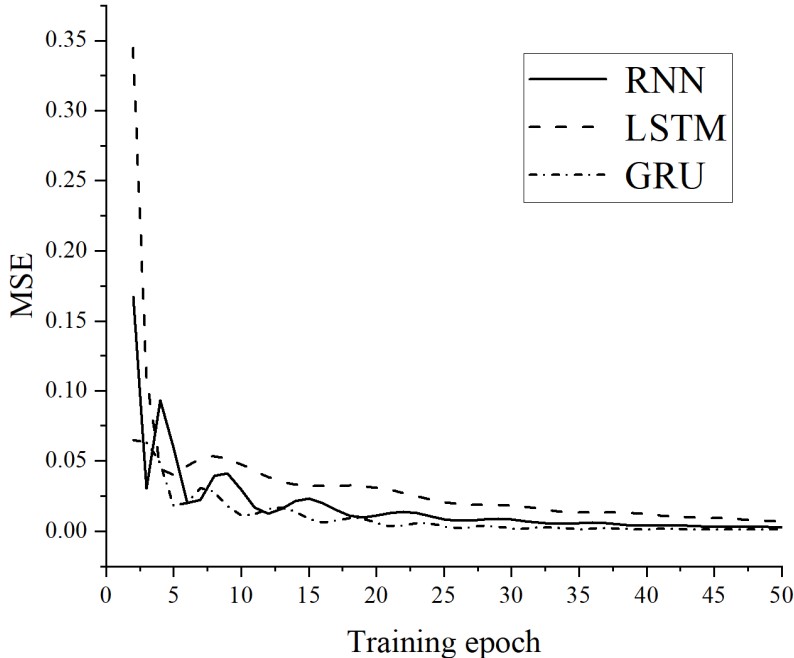

**Figure 4.** The training process of deep learning models.

In the test dataset, the MAE of the three models is small, and the MAPE of LSTM and the GRU is less than 1%. The $R^2$ values are greater than 0.99, indicating that there is a significant linear relationship between the true value and the predicted value, which can be obtained in Table 3.

**Table 3.** Table of prediction effect of DL.

| Models and Indicators | RNN | LSTM | GRU |
|:---:|:---:|:---:|:---:|
| MAE | $1.50 \times 10^{-2}$ | $5.18 \times 10^{-4}$ | $2.16 \times 10^{-4}$ |
| MAPE | 4.56 | 0.16 | 0.05 |
| $R^2$ | 0.99 | 0.99 | 0.99 |

Typical stations, Yancheng station, Guiyang station and Beijing station, are selected from 23 stations, and prediction analysis is performed. Figure 5 shows that the three deep learning models can better simulate the actual local data of the initial flowering period. The fluctuation trend of the LSTM and GRU models is different from that of the actual extreme years, which is mainly characterized by hysteresis, and the simulated fluctuation change is always smaller than the actual value in the year with obvious changes.

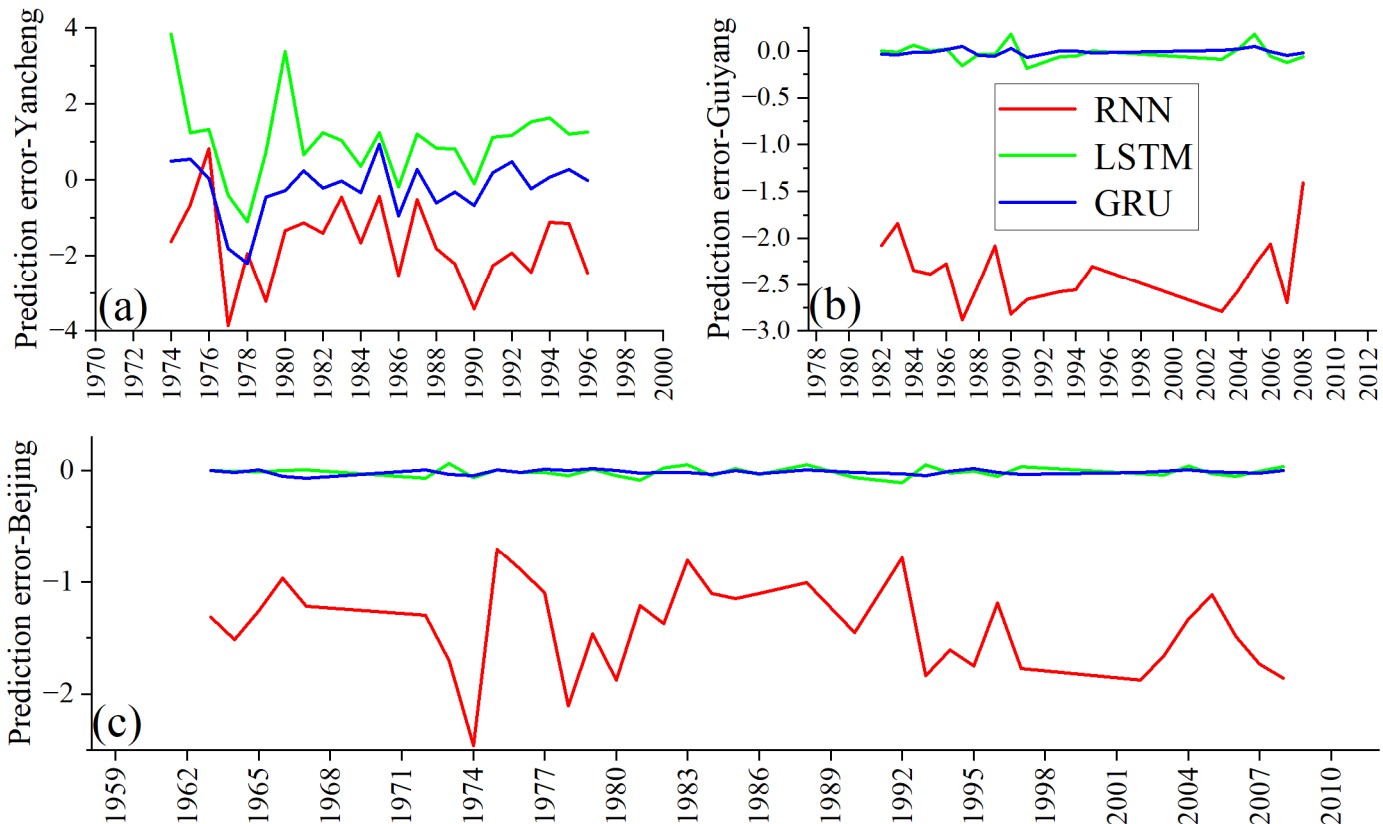

**Figure 5.** Interannual variation figure of prediction error of (**a**) Yancheng city, (**b**) Guiyang city, and (**c**) Beijing city.

### 3.3. Interpretability of DL Models

In Deep SHAP, a single sample will output SHAP values of different factors. We used the data of all samples including the training dataset and test dataset, which is a matrix of 89 meteorological factors and ordinal number of initial flowering period. Therefore, a matrix of SHAP values of the same size can be obtained. We explore the importance of different meteorological factors to model prediction by taking the average SHAP value of the whole sample as the factor contribution rate and analyze the stability of different factors

by using the change in the SHAP value of different samples in various meteorological factors as the stability index.

Figure 6a–c shows the analysis thermodynamic diagrams of RNN, LSTM and GRU. Its *x*-axis represents 89 meteorological factors, which are shown by x1-x89, and the order of meteorological factors is from the effective accumulated temperature of 0 °C to early average pressure for ten days in April. The *y*-axis represents 357 samples, which are shown by A1-A357. According to SHAP, the value greater than 0 in the thermodynamic diagrams promotes the prediction effect of the model, while the value less than 0 reduces the prediction effect, and we used color palette to indicate whether the value is greater than 0.

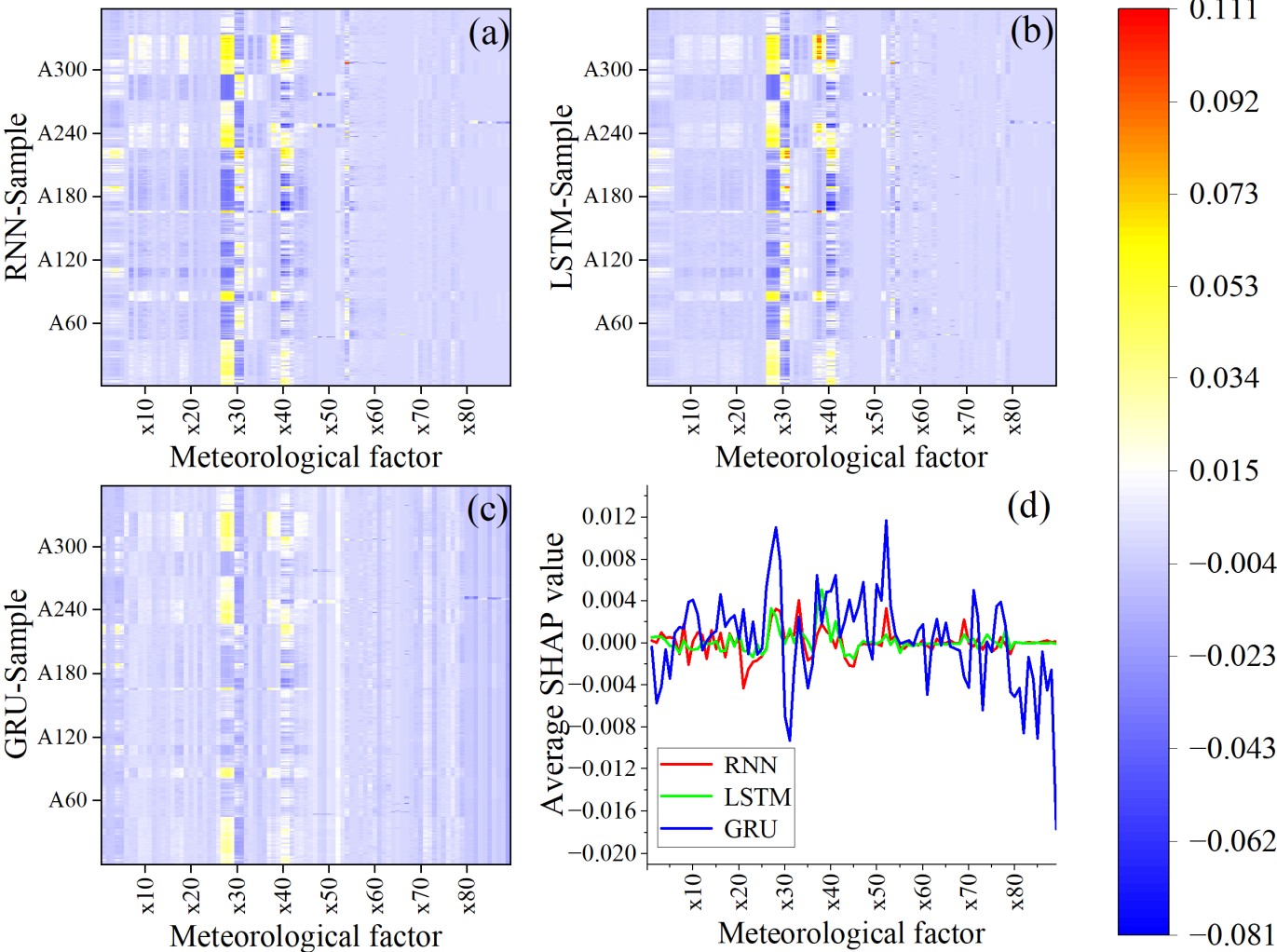

**Figure 6.** SHAP analysis figure of deep learning models (**a**). RNN contribution analysis thermodynamic diagram; (**b**). LSTM contribution analysis thermodynamic diagram; (**c**). GRU contribution analysis thermodynamic diagram; (**d**). Analysis figure of average contribution of each factor.

Therefore, Figure 6a–c can reflect the contribution rate stability of each meteorological factor though the change of SHAP value of each factor in different samples. In the thermal image, the meteorological factors with obvious fluctuations in the contribution rates of different models are similar and mainly concentrated in various factors related to the minimum temperature. The SHAP value of temperature factors is stable near the positive value, while the SHAP value of pressure factors is stable at the negative value.

According to Figure 6d, among different deep learning models, the average factor contribution rate is different with the range being 0.011 and normalized standard deviation

being 0.2, but in general, temperature factors are more important to the model with 58.6% of temperature factors values being higher than 0. Other factors are less important to the model, and some factors have negative SHAP values, which means they have a negative role in improving model prediction. GRU is more sensitive to input factors, so the absolute value of the contribution rate of GRU factors is higher than that of the other two models, while LSTM is the least sensitive to input factors, among which the absolute value of contribution rate of RNN, LSTM and GRU are $8.22 \times 10^{-4}$, $5.87 \times 10^{-4}$, $3.25 \times 10^{-3}$.

### 3.4. Comparison between DL and the Traditional Prediction Model

Non-DL flowering prediction methods usually use a few meteorological factors to establish regression models to forecast the initial flowering period, such as a multiple linear regression model, which is a linear regression model with multiple independent variables [38,39]. However, the simple linear models have difficulty accurately predicting flowering period. Chen and others have established a linear mode of multiple linear regression and nonlinear models of polynomial regression between the cherry flowering period and climate factors, and determined that they have a good simulation effect for the nonlinear modes with an average error of prediction less than 1.5 d [23,40]. In the neural network structure of deep learning, there are linear operations such as the convolution layer and nonlinear operations such as the activation function. To test the prediction effect of the deep learning model, we also select the multivariate linear regression model based on the accumulated temperature as the contrast for comparison.

According to the research of most scholars [9,13,19,20,22,23,33], we use the effective accumulated temperature (whose lower limit temperatures are 0 °C, 3 °C, 5 °C and 10 °C), active accumulated temperature and total accumulated temperature as variable factors to establish a multiple linear regression model:

$$y = 166.33x_1 - 261.86x_2 + 59.07x_3 + 38.79x_4 - 0.85x_5 - 1.57x_6 + 0.77 \tag{11}$$

where $x_1$, $x_2$, $x_3$, and $x_4$ are the effective accumulated temperatures whose lower limit temperature are 0 °C, 3 °C, 5 °C and 10 °C, $x_5$ is the active accumulated temperature, and $x_6$ is the total accumulated temperature. The coefficient of each independent variable is its linear relationship with $y$.

According to the deep learning models and multiple linear regression model, the prediction accuracy of each model is evaluated via MAE, MAPE and $R^2$, and the results can be obtained from Table 4.

**Table 4.** Table of comparison between deep learning model and multiple linear regression.

| Model / Indicator | Deep Learning Model | | | | Multiple Linear Regression Model |
|---|---|---|---|---|---|
| | **RNN** | **LSTM** | **GRU** | **Mean** | |
| MAE | $1.50 \times 10^{-2}$ | $5.18 \times 10^{-4}$ | $2.16 \times 10^{-4}$ | $5.12 \times 10^{-3}$ | 0.06 |
| MAPE | 4.56 | 0.16 | 0.053 | 1.59 | 15.45 |
| $R^2$ | 0.99 | 0.99 | 0.99 | 0.99 | 0.84 |

By comparison, the accuracy of the deep learning model was significantly higher than that of the multiple linear regression model with a confidence level of 0.05. And through the multicollinearity analysis, it can be found that in the multiple linear regression model, there is a collinearity problem between the 0 °C effective accumulated temperature and the 10 °C effective accumulated temperature.

### 3.5. Spatial Distribution and Interpolation of Prediction for DL

Due to the relationship between meteorological elements and space (longitude and latitude), we utilized all phenological data and verified the deep learning model at different stations to show the impact of spatial factors on flowering prediction. We import all the samples into the trained models and calculate the difference with the true value to get

the prediction error. When the error is more than 0 d, it means that the prediction results are ahead of the initial flowering period. The smaller the absolute error, the better the prediction effect. According to the Figure 7, the prediction average error of the RNN model lag behind the true value, mainly focusing on (−2d, −1d) and (−3d, −2d), while the prediction error of LSTM and GRU are mainly focused on (−1d, 0d), but the error of LSTM results exceeds 3d. By comparison, the prediction results of the GRU model are more accurate and stable.

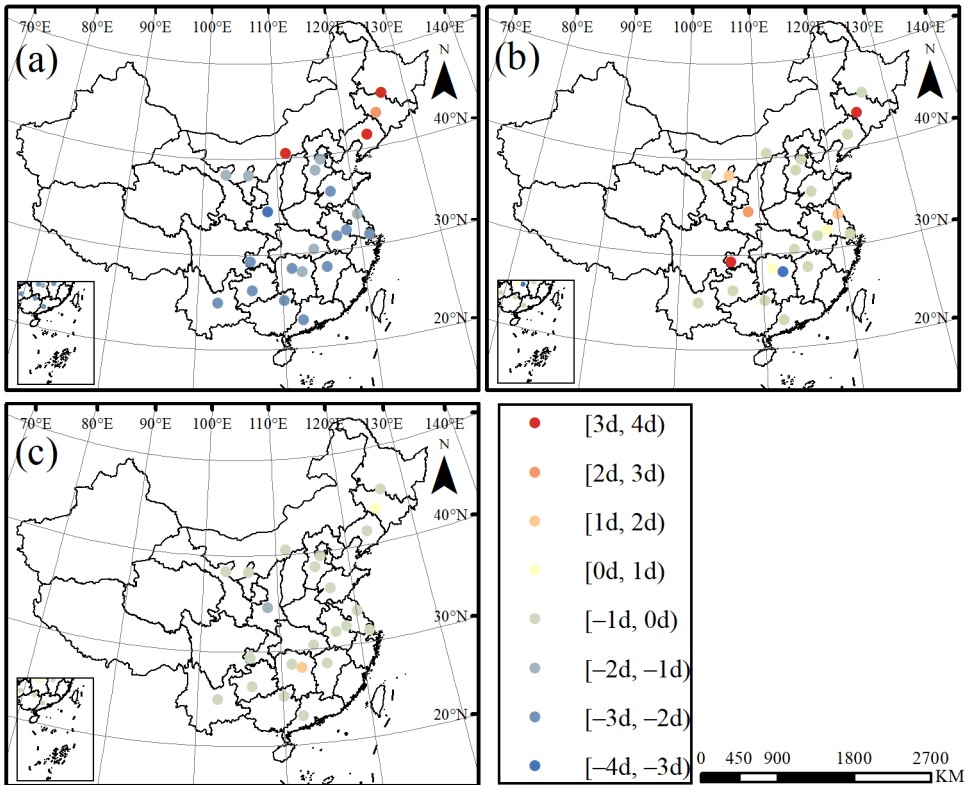

**Figure 7.** Spatial distribution map of (**a**) RNN, (**b**) LSTM, and (**c**) GRU prediction error.

Since most phenological observation stations in the dataset are concentrated in major urban areas of China and observation data in Northwest and Southwest China are missing, inverse distance weighting (IDW) is used for the average spatial prediction results of deep learning models. According to Figure 8, the interpolation results of the three models show similar characteristics. The similar characteristics are that in terms of latitude, ordinal number of initial flowering period gradually increases from low latitudes 15° N to high latitudes 55° N and present an obvious hierarchical structure, which is the layered structure of early, middle and late initial flowering periods from south China to north. The late flowering area mainly consists of Inner Mongolia and the three eastern provinces of Heilongjiang, Jilin and Liaoning, the middle flowering area mainly consists of central China, and the early flowering area mainly consists of the Yangtze River Delta, including Jiangsu, Zhejiang and Shanghai. The early flowering area and late flowering area have obvious differences in the initial flowering period. A possible main reason is that the late flowering area has a higher latitude, a smaller solar altitude angle, and less radiation, so the accumulated temperature and other resources are insufficient.

The prediction ordinal number of initial flowering period in different regions is similar in the three DL models with an average leaner trend between prediction value and years being −0.01, which means that the initial flowering period of *P. orientalis* in China will advance by about 1.31d each year.

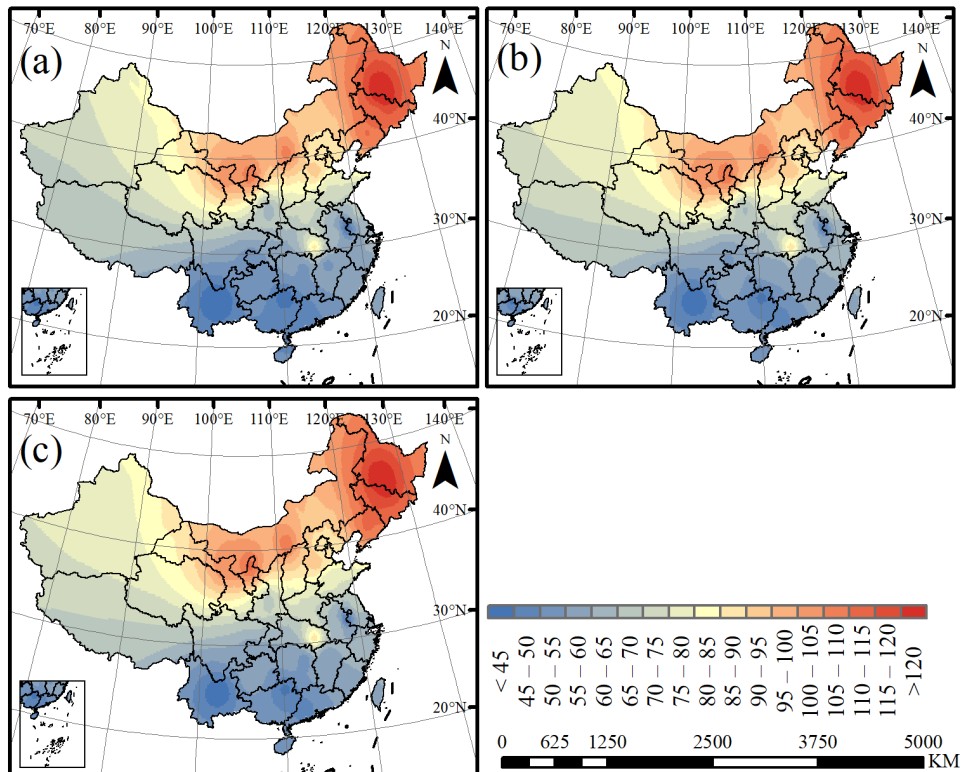

**Figure 8.** Spatial interpolation map of (**a**) RNN, (**b**) LSTM, and (**c**) GRU's average prediction.

## 4. Discussion

In this research, we employed deep learning to excavate the deep information relationship between the initial flowering period and phenology and realized a long-term flowering prediction model in China. The accuracy of the RNN, LSTM and GRU deep learning models is significantly higher than that of the traditional flowering prediction models based on multiple linear regression. Via interpretability analysis and spatial analysis, model stability problems such as factor sensitivity and error spatial distribution are explained.

From the viewpoint of some scholars, temperature is the main influencing factor affecting phenology [41–47], because it acts as a signal to regulate the dormancy process of plants [48]. Therefore, the mathematical regression models are built around accumulated air temperature, average air temperature and other factors related to temperature such as effective accumulated air temperature, etc. However, such models may have errors in the prediction effect over a short time and couldn't be applied to nationwide initial flowering period forecasts with the MAE, MAPE being higher than DL models and $R^2$ being lower.

Due to different meteorological conditions, the flowering period presents diversity in space [30]. In addition, because of the impact of climate change, the change of meteorological conditions in China is also different over time, showing the increase of annual temperature and precipitation [49,50]. This research achieves accurate nationwide prediction of a single species in China with the error of the initial flowering period reduced to less than 1 d, which provides more accurate data support for phenology research. With the development of industrialization, carbon emission might be the main factor affecting the opening process of flowers. Thus, this research provides a model basis for quantitative research on flowering changes in future scenarios.

However, there are some uncertainties in this research. The first uncertainty pertains to the data. We use the data of observation stations in major cities in China, with missing data from western China, which causes serious deviations between the prediction results and the actual value in this region. For DL, the SHAP of different models varies, and there is an obvious difference in the contribution of the three models to some meteorological factors, which makes it difficult to judge the correlation between such factors and the flowering period.

## 5. Conclusions

We predicted and analyzed the initial flowering period of *P. orientalis* in China through DL model, and the most important results of our study can be summed up as follows:

(1) The initial flowering in China mainly occurs from the beginning of February to the end of April, and it has spatial differences, which are later in northern China than in southern China.

(2) The DL model is suitable for nationwide flowering prediction in China, and the average error of DL is only within 2 d.

(3) Comparing the RNN, LSTM and the GRU, we find that the GRU is more suitable for the prediction model of initial flowering, with higher accuracy and more stable spatial predictions.

(4) The initial flowering period of *P. orientalis* in China presents obvious hierarchical characteristics, which are mainly manifested in the southern region where the flowering period is the earliest. With the increase in latitude, the initial flowering period gradually increases from south to north.

Although the variation in the contribution degree of output in the prediction of the initial flowering period can suggest different mechanisms of meteorological disasters affecting flowering process, our research is still insufficient.

**Author Contributions:** G.J.: visualization and writing. X.Z.: conceptualization and supervision. X.S., Y.S. and K.Y.: data collection and writing. W.S.: software and formal analysis. All authors have read and agreed to the published version of the manuscript.

**Funding:** This research was funded by the National Natural Science Foundation of China Project (41805049), National Students' Platform for Innovation and Entrepreneurship Training Program (202210300060Z) and NUIST Students' Platform for Innovation and Entrepreneurship Training Program (XJDC202210300493).

**Institutional Review Board Statement:** Not applicable.

**Informed Consent Statement:** Not applicable.

**Data Availability Statement:** Not applicable.

**Acknowledgments:** We thank the National Earth System Science Data Center, the Earth Big Data Science Data Center of the Chinese Academy of Sciences and the China Meteorological Science Data Sharing Network for data support. We acknowledge the High Performance Computing Center of Nanjing University of Information Science and Technology for their support of this work.

**Conflicts of Interest:** The authors declare no conflict of interest.

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
