# Peer review of "Utility of Deep Learning Algorithms in Initial Flowering Period Prediction Models"

_agriculture, doi:10.3390/agriculture12122161_

Round 1
Reviewer 1 Report
Review of the submitted manuscript entitled Research on a Prediction Model of the Initial Flowering Period of Arborvitae Based on Deep Learning Algorithms
I read the submitted manuscript with great interest because of its interesting approach to modelling phenology. And indeed, the manuscript is very interesting, but not everything is clear to me. Specifically, the question is on which species or species the study was conducted. What is Arborivitae? When I type the species of the genus Thuja, mainly Thuja plicata and T. occidentalis, into a web browser is displayed. It is necessary to clearly define the object of study - the species or species using their Latin scientific names. What was the phenological data? How was the beginning of flowering defined? Why was the offset of flowering and its length not also attempted?
L51-61: Long-term phenological observation series are rarely available and limit such studies (small number of locations, a limited number of observed species, taxonomic bias, etc.).
Instead, flowering dates can be modelled using citizen science data (e.g., iNaturalist), bioclimatic maps and daily gridded climatic data (e.g. E-OBS). This way, flowering dates (onset, duration, and offset) were mapped for one of the widely distributed European spring plants for current and future climatic conditions.
Puchałka R. et al. (2022) Citizen science helps predictions of climate change impact on flowering phenology: A study on Anemone nemorosa. Agric For Meteorol 325:109133. https://doi.org/10.1016/j.agrformet.2022.109133
Please discuss it.
I will add more comments in the next round of review when the authors have more clearly defined what they studied.
Sincerely yours
Author Response
Thank you very much for reviewing the manuscript for us and putting forward useful suggestions for the manuscript. We also sincerely appreciate that because of your recognition of our work, we have revised the manuscript according to your comments, and hope our manuscript is well appropriate for Agriculture.
- What is Arborivitae? When I type the species of the genus Thuja, mainly Thuja plicata and T. occidentalis, into a web browser is displayed. It is necessary to clearly define the object of study - the species or species using their Latin scientific names.
Response: Thanks for the reviewer’s comment, it helps us to clearly define the object of study, so that the object can be more clearly known by the readers. The word we used are not Latin scientific name, so we have revised the Arborivitae throughout the manuscript to Platycladus orientalis to ensure a clear definition which is the specie we studied. (Line 95-104)
- What was the phenological data? How was the beginning of flowering defined?
Response: Thanks for the reviewer’s comment to help us fill in the missing definitons. We didn’t give clear definitions of these terms in the manuscript before. Phenological data are observational data that reflect periodic biological phenomena, which comes from the definition of phenology. Therefore, the phenological data of plants include the initial flowering period, the length of the flowering period, etc., and we only selected the initial flowering period for prediction research. According to Chinese phenological observation norms, the initial flowering period refers to the observation trees with the petals of one or several flowers fully open. Now we have added these definitions to the manuscript. (Line 132-134)
- Why was the offset of flowering and its length not also attempted?
Response: Thanks for the reviewer’s comment. At present, the flowering forecast work carried out by meteorological bureaus in various parts of China mostly revolves around the timing of the initial flowering period. And we hope to replace local forecast models with new algorithms which can be applied nationwide to avoid duplication of flowering forecasting work. Therefore, we started our research only on the prediction of the initial flowering period. We strongly agree with the reviewer that we will consider flowering length prediction in the next large-scale research.
- Long-term phenological observation series are rarely available and limit such studies.
Response: Thanks for the reviewer’s comment. We appreciate reviewer’s support and agree that this research should be based on more data. However, the area we studied was the Chinese region in order to address issues such as repeated flowering forecasts in China. Due to late collection of phenological data in China, the data are less and not fully public, and our research was based on available observations of phenological stations. In the next research, we will expand the study area to Europe and North America, because there are more public flowering data, and reviewer’s suggestions will be very helpful for our next research.
Thanks again for your suggestions. We look forward to working with you to move the manuscript closer to publication in the Agriculture.
Sincerely yours
Reviewer 2 Report
The work concerned forecasting the initial of thuja flowering in China on the basis of meteorological conditions with the use of various statistical tools: RNN (recurrent neural network), LSTM (long short-term memory), GRU (gated recurrent unit). The constructed models were assessed on the basis of several indicators: MAE, MAPE, R2.
Unfortunately, only phenological data from 23 stations in the years 1963-2015 were used to build the models. There was no information about the course of the flowering of thuja in the western and northern parts of China. At least every major physico-geographic land of China should be represented by 1 phenological station. In addition, the series of phenological data was not complete and therefore some years were overrepresented (Fig. 4). The data came from municipal stations, where the meteorological conditions differ from those outside the city. Can models built for urban areas be used to predict the initial of thuja flowering throughout China? The paper does not provide information on the performance of phenological observations. Also the paper not described conditions for the growth and development of thuja.
Apart from those mentioned above, I have many other doubts. What is average precipitation (table 1)? On what basis were the threshold values of the sum of the effective temperature selected: 0℃, 3℃, 5℃, 10℃. Why is the MSE (mean squared error) indicator described in the methodology? After all, the authors did not provide the MSE value in any table. Why were the values of the indicators assessing the training effect provided with an accuracy of up to 6 decimal places; e.g. R2 = 0.993931 (tab. 3-4)? How was the collinearity between the meteorological variables assessed in model 8 (page 11)? Were the regression coefficients of the model 8 statistically significant? How was the trend of the phenological appearance included in the models?
Author Response
Thank you very much for your key suggestions on our manuscript. We have revised the manuscript according to your suggestions, and hope our manuscript is well appropriate for Agriculture.
- Unfortunately, only phenological data from 23 stations in the years 1963-2015 were used to build the models. There was no information about the course of the flowering of thuja in the western and northern parts of China. At least every major physico-geographic land of China should be represented by 1 phenological station.
Response: Thanks for the reviewer’s comment. We agree with the reviewer. However, it’s a pity that the regional phenological data in China is not completely public, we have gone as far as possible to obtain phynological station data for major physico-geographic lands of China, and we selected thuja as our research object based on the published phenological data in China because the amount of data on this species is one of the highest among the 23 phenology stations. Therefore, we cannot guarantee that the prediction is good for all cities, especially in the northwest of China. And when we carried out spatial analysis, we didn’t use the DL method, although we have sufficient meteorological stations data, but used the IDW, because we have proved that the initial flowering period has a clear correlation with meteorological data based on 23 observation stations, and meteorological elements are spatially more used IDW for interpolation, so the final image we get is based on GIS, not directly using deep learning, which is mainly to visually represent the zoning characteristics of the timing of the initial flowering period of thuja in China.
- The series of phenological data was not complete and therefore some years were overrepresented.
Response: Thanks for the reviewer’s comment. We agree with the reviewer and we have taken some measures to try to avoid this problem. Due to the different start times of different phenological observation stations, their series are not exactly the same. In order to avoid the problem of that some years were overrepresented, we only consider the meteorological conditions when designing the models. In the same year, the spatial difference of meteorology is the most important, and in different times, the time difference of meteorology is also important, which avoids excessive consideration of the year.
- The data came from municipal stations, where the meteorological conditions differ from those outside the city. Can models built for urban areas be used to predict the initial of thuja flowering throughout China?
Response: Thanks for the reviewer’s comment. Thanks for the consideration and discussion in the regard. In our models, we only used meteorological factors as inputs, so the meteorological differences between different stations directly determine the predicted timing. In the areas that outside the cities, the meteorological conditions are different, so we can predict the flowering period in these areas by comparing the differences in meteorological factors. This is not verified, because flowering forecasts in no-urban areas are not mainstream, and we lack relevant observational data. (Line 289)
- The paper does not provide information on the performance of phenological observations.
Response: Thanks for the reviewer’s comment. We would love to ba able to show readers the information on the performance of phenological observations, but we are sorry to say that due to the limited sharing of phenological observation data in China, we cannot public the data in form of reanalysis datasets and other forms, so we only listed the observation initial flowering period of each station for statistical analysis (Table 2), not the specific flowering period of all stations. (Line 289)
- The paper not described conditions for the growth and development of thuja.
Response: Thanks for the reviewer’s comment. We have added a topic of the description of thuja into manuscript. (Line 101-104)
- What is average precipitation (table 1)?
Response: Thanks for the reviewer’s comment. In our research, the “average precipitation” is average precipitation from January to April that is a value and “average monthly precipitation from January to April” is monthly precipitation that are four values to one piece of data. Because precipitation is often an important meteorological condition in phenology research, in addition to the average precipitation, we also need more accurate average monthly precipitation. (Line 172)
- On what basis were the threshold values of the sum of the effective temperature selected: 0℃, 3℃, 5℃, 10℃.
Response: Thanks for the reviewer’s comment. We chose thresholds based on some knowledge of ecology. There are five major temperature zones in China, 5 °C is the threshold for temperate plants, 10 °C is the threshold for subtropical plants, because China has a wide range of temperate zones, so between the threshold of temperate and cold temperate zones, we chose 3 °C as the threshold temperature. In order to highlight the damage effect of high temperature and low temperature, we chose 40 °C and 0 °C as the threshold temperature in the ground temperature. In addition to accumulated temperature, more active accumulated temperature is used. And we strongly agree with the reviewer, so we have added the definition to the paper. (Page 3, Line 112 – Line 117; Page 5, Line 167 – Line 168; Page 12, Line 357 – Line 360)
- Why is the MSE (mean squared error) indicator described in the methodology? After all, the authors did not provide the MSE value in any table.
Response: Thanks for the reviewer’s comment. We used MSE as the loss function when training the DL models. The change of MSE with the epoch can be got in the Fig 4. (Line 296-297)
- Why were the values of the indicators assessing the training effect provided with an accuracy of up to 6 decimal places; e.g. R2 = 0.993931 (tab. 3-4)?
Response: Thanks for the reviewer’s comment. The reviewer’s suggestion provides constructive help for us to modify the decimal places. Our original idea was to distinguish the R2 of GRU and LSTM by 6 decimal places, but this is not necessary and does not meet the general requirements. So we changed the decimal places in Table 3-4 to 4 decimal places. Thanks again to the reviewer for this suggestion. (Page 10, Line 301; Page 13, Line 367)
- How was the collinearity between the meteorological variables assessed in model 8 (page 11)?
Response: Thanks for the reviewer’s comment. We import VIF to measure the collinearity problem in multiple linear regression, and obtain that the effective accumulated temperature of 0 ℃ and the effective accumulated temperature of 10 ℃ have collinearity. And we have added this into the manuscript. (Line 361-364)
- Were the regression coefficients of the model 8 statistically significant?
Response: Thanks for the reviewer’s comment. Different coefficients represent the linear relationship between different factors and y. We have added relevant explanations in the manuscript (Line 361-364)
- How was the trend of the phenological appearance included in the models?
Response: Thanks for the reviewer’s comment. Since the predicted value of the model is very close to the true value, we didn’t explain it too much. And we have added the analysis of flowering time to the Spatial distribution and interpolation of prediction for DL. (Line 402-405)
Thanks again for your suggestions. We look forward to working with you to move the manuscript closer to publication in the Agriculture.
Sincerely yours
Reviewer 3 Report
Dear,
The article entitled: Research on a Prediction Model of the Initial Flowering Period of Arborvitae Based on Deep Learning Algorithms refers to the application of deep learning techniques with meteorological data and the beginning of flowering of Arborvitae. These techniques are being widely applied in various fields and scientific research. The research, although not innovative, has scientific merit. The text needs to be more explained, detailing a little more the methodology, the results and the discussions with more care and depth. Some parts of the text need to be rewritten as follows:
Lines 15 to 20. Based on the daily meteorological data ... Too long sentence! Improve the writing! Use punctuation correctly!
Line 24 - ...based on a accumulated temperature... What type of temperature? of the air? of the soil? of the vegetation?
Line 25 - ..., the temperature type factor has the highest contribution rate...What is the value of this high contribution rate?
Lines 26 and 27 - The stability of the contribution rate of the factors related to the daily minimum temperature factor has obvious fluctuations. Explain why the stability of the contribution rate of the factors related to the daily minimum temperature factor has obvious fluctuations? Write it in the text! What are the factors related to air temperature? Write them down. Be objective and clear!
Line 30: Keywords: Arborvitae,. The keywords Arbonitae, Initial flowering period, Deep learning algorithm are contained in the title. The word Meteorology is too generic. The keywords have to be different from the words in the title. Suggestion: look in the abstract or methodology for words that are important to identify your paper! Suggestion: recurrent neural network; IDW; Cumulative air temperature;
Line 41. ..., and temperture is the main factor... Specify what kind of temperature? Air temperature?
Line 42. Important progress has been made in phenological research... Exemplify the type of progress that has been made? Write in the text.
Line 47 - Give the meaning of ASYMCUR GDH. The chill unit model has appeared other times in the text. Could you add in the text a short explanation of how this model works!
Line 50. years of phenological and meteorological data... Specify the types of data used? flowering dates? air temperature data? Relative humidity? solar radiation? You write in such a way that the reader is left in doubt in your statements! Avoid this by explaining the referring terms!
Line 51. ... through modelling (?). Write the type of modelling referred to by Hakkinen and co-workers.
Line 57-58. ... the analysis of local meteorological mechanisms (?). Cite examples of these mechanisms. If you don't, we won't know what mechanism you were thinking of when you wrote it, because there are so many!
Lines 58-59 - In 2019, Wu D et al. [22] conducted applicability analysis through the forecasting model... Explain further what applicability analysis is? Give examples. What type of forecating model was applied? Write it down!
Line 60 - ... the main impact factors(???) which?
Line 62 -63 - Explain better what is this ... limited effective range? ...ambiguous impact factors?
Line 64. ... different plants (????) which ones?
Line 66 - ... ... wide demand for flowering forecasting (??) Give examples of agricultural crops in China.
Line 78 Include a description of the study area: climate (air temperature, solar radiation, precipitation), soil, crop characteristics and crop requirements: flowering period, amount of water, temperature. Economic importance for the country of the agricultural crop to be studied. Then write about obtaining the phenological and meteorological data.
Line 79 - 85 ... Very long sentences. Cite Figure 1 in the text.
... we collected observation data of the initial flowering of arborvitae... What are these data? Dates of the initial flowering? You should write one topic about the flowering data and another about the meteorological data that will be used in the research. What period of weather data is used?
Line 94 - In Figure 1 add north!
Line 96 Methods. Here you detail the methods that will be applied. I did not see you write about multiple linear regression. However, they appear in the results, why? Detail in the methodology the multiple linear regression model!
Line 98. The effect of temperature (?)... air temperature? Specify the type of temperature.
Line 101-102. ... the accumulated temperature or degree days ?
Line 103. ... the lower limit temperature or lower basal temeperature? I don't know what kind of temperature you mean! Improve it!
Line 105. ... the effective accumulated temperature (Quote?). Insert the Equation in the text. For example, the effective accumulated temperature (Citation?) expressed by Equation 1. Is this effective accumulated temperature of air? Write it down!
Line 110. ...0 °C, 3 °C, 5 °C, and 10 °C take the space between the number and °C.
Line 113. ...other major factors (?) add examples of factors and the citation!
Line 114. table of meteorological. Add the units of measurement for rainfall (?) relative humidity ( ?), pressure (?). Explain how you found the number of factors, because the meteorological factors for temperature are 7 and the number of factors 46? I don't understand! Where did they come from, i.e. the 23 weather stations?
Explain in the text how these factors will be used? And what for? How were the gaps in the meteorological data treated? The application of this data seems obvious to you, but for the reader it can be quite complicated to understand how this data will be used in the bloom prediction model, so better explain it clearly in the text!
Line 118 With each meteorological factor (Table 1)... add Table 1.
.... and Julian day (initial flowering date?)... How did you establish Julian day? Write.
How are the data normalized? Explain in the text!
Lines 120-121 At a ratio of 7:3, we divided the training dataset and test dataset for model training and model effect evaluation. Explain this 7:3 ratio better? Detail in the text.
Lines 124-125. ... a better fitting ability. Write examples in the text.
...Compared with other common networks (which ones?)...
Line 139. ... function (Equation 2)
Lines 153 - 155 ,,,The mean squared error (MSE, Equation 3)... (MAE, Equation 4),... (MAPE, Equation 5)...(coefficient of determination (R², Equation 6) write the parameters and their corresponding equation!
Line 159. ...will be more accurate. Write the value of the precission!
Line 161 and 162 - R2 change to R².
Line 167. game theory (reference?). Give a short explanation of the game theory in the text.
Line 174 - ...flowering period (Reference?) expressed by Equation 7: Add equation 7 in the text.
Line 196. ...analysis thermodynamic diagram and analysis chart (Reference?)...
Line 197. ...the average contribution of each factor (Figure 2)
Line 199..... Figure 2 should be presented right after a description of the study area. Then each step of the flowchart would be presented.
Line 204 ... the Julian day of cities in northeast China a is nearly 80 d... Improve the writing of this sentence! What do you mean by ... the Julian day of cities in northeast China (cities?) a is nearly 80 d... 80 d = 80 days? Yes.
line 205 ... coastal cities in south China (Which ones? Name them.)
Line 209. ... station is 23.93 (add days) ...the minimum of Harbin station is 1.50 (Add days).
Line 211- There are obvious interannual fluctuations and spatial differences... Besides latitude, what other factors affect flowering at these sites?
Lines 212 -213. ...the degree of dispersion is large (How much?)
Line 215. Improve the writing of value/d in table 2 by value (d).
Line 217. With an increase in the training epoch... Explain better what the training time means.
Line 229. Figure 5 shows that... Isn't it Figure 4?
Lines 238-242 - How many sample factors were used? These samples refer to which data and locations? Write in the text.
Line 243. Figure 5 (a), (b) and (c) reflect the contribution rate stability of each factor in different samples. Specify the number of samples used! Improve the explanation of Figure 5. It is confusing! What does each axis represent? What does the color palette mean? The results are good, but the explanation needs more work to show the importance of the technique. One way is to make the figures more detailed and explore the results more.
Line 246. What are the values of the contribution rates of the models?
Line 250-251 .... the average factor contribution rate is different (?) How much is this difference?
... temperature factors are more important to the mode? Why? How did you get this result? From figure 5?
Line 254. ... GRU factors is greater than... Greater than how much?
Line 260. Make it clear what A1 to A37 are? x1-x89? writing that they are samples and weather factors does not justify it! What are these factors? What are these samples? From which cities?
Line 262 - 265: The traditional prediction model of the flowering period is often selected for prediction by establishing a multiple regression prediction model... The sentence is very confusing! Rewrite it! What is SPSS? What degree of influence of various factors is this? What factors are these? Very badly written! Multiple linear regression? Explain this in the methodology!
Line 265 - However, there is an upper limit for the prediction accuracy of simple linear models [28,29]. What is the value of the upper limit for the prediction accuracy? Put it in the text!
Line 266 - Chen and others have established linear and nonlinear models between the cherry flowering period and climate factors (Which ones?) and determined that they have a good simulation effect (What effect 'is that?) for the nonlinear mode (?). Very confusing sentence! Rewrite it!
Line 268- 270: In the neural network structure of deep learning, there are linear operations (Which?) with a convolution layer as the core and nonlinear operations (Which?) with an activation function (?) as the core. Rewrite!
Line 273. According to the research of most scholars (Cite them)...we use the effective accumulated temperature of ≥0℃, ≥3℃, ≥5℃, and ≥10℃. Now that's what's hard to understand! What are these confusing variables? Rewrite!
Line 276. x1 >= 0°C, x2 >=3°C... You accumulate this to infinity! I can't understand this! Rewrite. The traditional prediction model of the flowering period is often selected for prediction by establishing a multiple regression prediction model,
Line 278. According to the deep learning models and multiple linear regression model (Table 4)... Include the table in the text.
Line 282. the accuracy of the deep learning model was significantly higher than... What is the confidence level to say it is significant? Put in the text!
Line 285. Due to the obvious spatial relationship of meteorological elements...With what is this spatial relationship obvious? Confusing! Write better! You use the word obvious a lot! Line
287. Cite Figure 6 in the text!
Lines 288-289. the prediction results of the RNN model lag behind the true value, mainly focusing on [-2d, -1d) and [-3d, -2d), while the prediction error of LSTM and GRU are mainly focused on [-1d, 0d), but the error of LSTM results exceeds 3d. Confusing writing! What is the true value? The date of bloom observation? What does the interval [-2d, -1d) mean? Explain in the text! What does -2d mean? Underestimated by 2 days? 0d is an error or a model hit?
Line 297. the three models show similar characteristics... What are these similar characteristics? Write in the text!
Line 298 - Julian days gradually increase from low latitudes to high latitudes. Would not be days of flowering? Julian day is equal to days occurring in a year. So, improve it there! Low latitudes to high latitudes (What are the low and high latitude values?) Write in the text. ...obvious hierarchical structure? What is this obvious hierarchical structure? What do you mean!
Lines 299-301 I don't know China, so please identify in Figure 7 Mongolia and the Yangtze River Delta? Also add on all maps the latitude and longitude and the direction from North!
Line 305 - 307 What is the type of humidity? What is the type of precipitation? Why are the distributions of air temperature and humidity uneven? What are the overarching factors that affect the flowering period? Explain in the text.
Line 314 -315. flowering prediction models... Give examples of the traditional flowering prediction model. These traditional flowering prediction models were not well explained throughout the text. Improve it!
Line 318-320. Therefore, the mathematical regression models are built around accumulated temperature, average temperature and other factors related to temperature. Specify if the temperature is of the canopy surface air! Name these other factors!
... models may have obvious error ? Cite the obvious errors!
Line 321. ... the traditional prediction model cannot realize the nationwide prediction of initial flowering. Explain why?
Line 326 - ... a better effect in the prediction of multiple species. What do you mean by this? Confusing! The model can only be applied to multiple species and not to a single species?
Line 328. ... the atmospheric environment ?
Author Response
Thank you very much for your comments. Your suggestions are very helpful for the revision of our article. We have improved our writing, and hope our manuscript is well appropriate for Agriculture.
- Lines 15 to 20. Too long sentence! Improve the writing! Use punctuation correctly!
Response: Thanks for the reviewer’s comment. We have improved the writing here. (Line 14-21)
- Line 24 …based on accumulated temperature… What type of temperature? Of the air? Of the soil? Of the vegetation?
Response: Thanks for the reviewer’s comment. The type of temperature of accumulated temperature is air temperature. And we strongly agree with the reviewer, so we have modified this definition in the paper. (Line 24)
- Line 25 - ..., the temperature type factor has the highest contribution rate...What is the value of this high contribution rate?
Response: Thanks for the reviewer’s comment. We have added the value of contribution rate to the manuscript. (line 25-27)
- Lines 26 and 27 - The stability of the contribution rate of the factors related to the daily minimum temperature factor has obvious fluctuations. Explain why the stability of the contribution rate of the factors related to the daily minimum temperature factor has obvious fluctuations? Write it in the text! What are the factors related to air temperature? Write them down. Be objective and clear!
Response: Thanks for the reviewer’s comment. We have added the data to explain the influence, and we have also provided more explanation to the manuscript. (Line 27-29)
- Line 30: Keywords: Arborvitae,. The keywords Arbonitae, Initial flowering period, Deep learning algorithm are contained in the title. The word Meteorology is too generic. The keywords have to be different from the words in the title. Suggestion: look in the abstract or methodology for words that are important to identify your paper! Suggestion: recurrent neural network; IDW; Cumulative air temperature;
Response: Thanks for the reviewer’s comment. We strongly agree with the reviewer’s suggestion, so we have modified this definition in the paper. (Line 33)
- Line 41. ..., and temperature is the main factor... Specify what kind of temperature? Air temperature?
Response: Thanks for the reviewer’s comment. The kind of the temperature is air temperature, and we have modified this definition in the paper. (Line 44)
- Line 42. Important progress has been made in phenological research... Exemplify the type of progress that has been made? Write in the text.
Response: Thanks for the reviewer’s comment. We have added the type of progress to the paper. (Page 1, Line 45; Page 2, Line 46 – Line 48)
- Line 47 - Give the meaning of ASYMCUR GDH. The chill unit model has appeared other times in the text. Could you add in the text a short explanation of how this model works!
Response: Thanks for the reviewer’s comment. We have added the explanation of ASYMCUR GDH model to the manuscript and explained how the chill unit model work here. (Page 2, Line 49 – Line 52; Page 2, Line 54 – Line 57)
- Line 50. years of phenological and meteorological data... Specify the types of data used? flowering dates? air temperature data? Relative humidity? solar radiation? You write in such a way that the reader is left in doubt in your statements! Avoid this by explaining the referring terms!
Response: Thanks for the reviewer’s comment. We have introduced the data types in this sentence, and given more explanations to the statements in the manuscript to avoid misunderstanding. (Line 58 – Line 60)
- Line 51. ... through modelling (?). Write the type of modelling referred to by Hakkinen and co-workers.
Response: Thanks for the reviewer’s comment. We have added the type of model to the manuscript. (Line 58 – Line 60)
- Line 57-58. ... the analysis of local meteorological mechanisms (?). Cite examples of these mechanisms. If you don't, we won't know what mechanism you were thinking of when you wrote it, because there are so many!
Response: Thanks for the reviewer’s comment. We have added the examples and modified the sentence to the manuscript. (Line 67 – Line 68)
- Lines 58-59 - In 2019, Wu D et al. [22] conducted applicability analysis through the forecasting model... Explain further what applicability analysis is? Give examples. What type of forecasting model was applied? Write it down!
Response: Thanks for the reviewer’s comment. We have added the destination to the paper. (Line 69 – Line 71)
- Line 60 - ... the main impact factors(???) which?
Response: Thanks for the reviewer’s comment. We have added the types of main impact factors and modified this sentence to avoid ambiguity. (Line 72 – Line 73)
- Line 62 -63 - Explain better what is this ... limited effective range? ...ambiguous impact factors?
Response: Thanks for the reviewer’s comment. We have explained these two definitions to help readers better understand them. (Line 75 – Line 77)
- Line 64. ... different plants (????) which ones?
Response: Thanks for the reviewer’s comment. What we mean here is that we want to express that for various types of plants, so we have modified this sentence to avoid ambiguity. (Line 77)
- Line 66 - ... ... wide demand for flowering forecasting (??) Give examples of agricultural crops in China.
Response: Thanks for the reviewer’s comment. We have added the examples of agricultural crops in China. (Line 80 – Line 81)
- Line 78 Include a description of the study area: climate (air temperature, solar radiation, precipitation), soil, crop characteristics and crop requirements: flowering period, amount of water, temperature. Economic importance for the country of the agricultural crop to be studied. Then write about obtaining the phenological and meteorological data.
Response: Thanks for the reviewer’s comment. We have added the topics about the research region, and added the growth conditions of Platycladus orientails. (Line 101 – Line 104)
- We collected observation data of the initial flowering of arborvitae... What are these data? Dates of the initial flowering? You should write one topic about the flowering data and another about the meteorological data that will be used in the research. What period of weather data is used?
Response: Thanks for the reviewer’s comment. We have added more explanations and adjusted the paragraph structure. Since our introduction to obtaining data is relatively small, it is difficult to divide it into two topics, but we have divided it into paragraphs according to different data types. We hope to solve the problem that the introduction of the two data is too centralized. And we are very grateful to the reviewer for this comment. (Page 3, Line 132 – Line 135; Page 4, Line 136 – Line 148)
- Line 94 - In Figure 1 add north!
Response: Thanks for the reviewer’s comment. We have added north compass to all figures, and added longitude and latitude to the grid of figures (Line 149)
- Line 96 Methods. Here you detail the methods that will be applied. I did not see you write about multiple linear regression. However, they appear in the results, why? Detail in the methodology the multiple linear regression model!
Response: Thanks for the reviewer’s comment. Multiple linear regression is one of the main methods of traditional flowering prediction. We compare the results based on DL and multiple linear regression to determine whether DL can be applied to flowering prediction. And we have added the introduction of multiple linear regression to the paper. (Line 266, Line 247-248)
- Line 98. The effect of temperature (?)... air temperature? Specify the type of temperature.
Response: Thanks for the reviewer’s comment. The type of this temperature is air temperature, and we have modified this sentence. (Line 153)
- Line 101-102. ... the accumulated temperature or degree days?
Response: Thanks for the reviewer’s comment. The sentence here is to introduce the role of accumulated temperature. We have modified the manuscript to improve writing. (Line 157 – Line 163)
- Line 103. ... the lower limit temperature or lower basal temperature? I don't know what kind of temperature you mean! Improve it!
Response: Thanks for the reviewer’s comment. We have improved the sentence and added formula definitions to help understand. (Line 158 – Line 160)
- Line 105. ... the effective accumulated temperature (Quote?). Insert the Equation in the text. For example, the effective accumulated temperature (Citation?) expressed by Equation 1. Is this effective accumulated temperature of air? Write it down!
Response: Thanks for the reviewer’s comment. Accumulated temperature can be applied to air temperature and ground temperature, which we have added to the paper. (Line 163)
- Line 110. ...0 °C, 3 °C, 5 °C, and 10 °C take the space between the number and °C.
Response: Thanks for the reviewer’s comment. We have improved it in the manuscript. (Line 167 – Line 168)
- Line 113. ...other major factors (?) add examples of factors and the citation!
Response: Thanks for the reviewer’s comment. This the other main factors refer to the meteorological factors we used, which can be got in Table 1, and we have modified this sentence. (Line 171-172)
- Line 114. table of meteorological. Add the units of measurement for rainfall (?) relative humidity (?), pressure (?). Explain how you found the number of factors, because the meteorological factors for temperature are 7 and the number of factors 46? I don't understand! Where did they come from, i.e. the 23 weather stations?
Response: Thanks for the reviewer’s comment. We have added the units in Table 1. Since the initial flowering period of Platycladus orientails is mainly in the middle of April, we focused on the meteorological data from January to April. Since some meteorological factors require higher time resolution, such as accumulated temperature, we have considered the influence of accumulated temperature decade-by-month, which is a factor for processing accumulated temperature every ten days, so it has more factors, and we have replace accumulated temperature decade-by-month with accumulated temperature for ten days to avoid ambiguity. For accumulated temperature, after reading the daily average temperature from the meteorological station, we calculate and obtain the accumulated temperature through equation 2, so we distinguish between meteorological data and meteorological factors. (Line 165-171, Line 173-178)
- Explain in the text how these factors will be used? And what for? How were the gaps in the meteorological data treated? The application of this data seems obvious to you, but for the reader it can be quite complicated to understand how this data will be used in the bloom prediction model, so better explain it clearly in the text!
Response: Thanks for the reviewer’s comment. We have added explanations on the manuscript. (Line 173-178, Line 292)
- Line 118 With each meteorological factor (Table 1)... add Table 1.
.... and Julian day (initial flowering date?)... How did you establish Julian day? Write.
Response: Thanks for the reviewer’s comment. We are very sorry for this. Julian day is a wrong use of the timing of the initial flowering period. We have made changes when we submitted the manuscript, but not completely. We are sorry for the misunderstanding, and we have made changes to this. (Line 182-183)
- How are the data normalized? Explain in the text!
Response: Thanks for the reviewer’s comment. We have added the equation of normalization to the manuscript. (Line 188-191)
- Lines 120-121 At a ratio of 7:3, we divided the training dataset and test dataset for model training and model effect evaluation. Explain this 7:3 ratio better? Detail in the text.
Response: Thanks for the reviewer’s comment. We have added the advantage of dividing to the manuscript. (Line 184-187)
- Lines 124-125. ... a better fitting ability. Write examples in the text.
Response: Thanks for the reviewer’s comment. We have given the example and added the citation to the manuscript. (Line 194-195)
- ...Compared with other common networks (which ones?)...
Response: Thanks for the reviewer’s comment. We have given the example to the manuscript. (Line 195-196)
- Line 139. ... function (Equation 2)
Response: Thanks for the reviewer’s comment. It may be a problem with the word version. We have reinserted this equation hoping that you can see successfully. Line (212-214)
- Lines 153 - 155 ,,,The mean squared error (MSE, Equation 3)... (MAE, Equation 4),... (MAPE, Equation 5)...(coefficient of determination (R², Equation 6) write the parameters and their corresponding equation!
Response: Thanks for the reviewer’s comment. We have reinserted the equations hoping that you can see successfully. (Line 156-157)
- Line 159. ...will be more accurate. Write the value of the precission!
Response: Thanks for the reviewer’s comment. About the advantages of MSE, we have added a reference and improved this sentence. (Line 232)
- Line 161 and 162 - R2 change to R².
Response: Thanks for the reviewer’s comment. We have changed the format of it. (Line 234)
- Line 167. game theory (reference?). Give a short explanation of the game theory in the text.
Response: Thanks for the reviewer’s comment. We have added the explanation of the game theory in the text. (Line 239-242)
- Line 174 - ...flowering period (Reference?) expressed by Equation 7: Add equation 7 in the text.
Response: Thanks for the reviewer’s comment. The equation here calculates the contribution rate. And we have improved the writing and reinserted the equation hoping that you can see successfully. (Line 244-248)
- Line 196. ...analysis thermodynamic diagram and analysis chart (Reference?)...
Response: Thanks for the reviewer’s comment. The thermodynamic diagram here is a visual method to help better analyze the contribution rate of each factor. And we have deleted this sentence. We have added the explanation to 3.3. (Line 317-326)
- Line 197. ...the average contribution of each factor (Figure 2)
Response: Thanks for the reviewer’s comment. In this section, we introduced too much about the method of interpretative analysis, so we have deleted this sentence.
- Line 199... Figure 2 should be presented right after a description of the study area. Then each step of the flowchart would be presented.
Response: Thanks for the reviewer’s comment. We agree with the reviewer and have adjusted the structure of the article. (Line 106-130)
- Line 204 ... the Julian day of cities in northeast China a is nearly 80 d... Improve the writing of this sentence! What do you mean by ... the Julian day of cities in northeast China (cities?) a is nearly 80 d... 80 d = 80 days? Yes.
Response: Thanks for the reviewer’s comment. We have improved this sentence, and we are sorry for the mistake here. (Line 277)
- line 205 ... coastal cities in south China (Which ones? Name them.)
Response: Thanks for the reviewer’s comment. We have added the name of coastal cities in south China to the manuscript. (Line 277-278)
- Line 209. ... station is 23.93 (add days) ...the minimum of Harbin station is 1.50 (Add days).
Response: Thanks for the reviewer’s comment. We have added the unit to the manuscript and the Table 2. (Line 283-284)
- Line 211- There are obvious interannual fluctuations and spatial differences... Besides latitude, what other factors affect flowering at these sites?
Response: Thanks for the reviewer’s comment. It can be seen from Table 2 that the flowering period is related to latitude, and the impacts caused by latitude are diverse and complex, the most important of which is to affect the weather, such as air temperature, precipitation etc. Similarly, longitude will also lead to differences in meteorological conditions in different regions. In our research, we only consider the relationship between meteorological factors and flowering period, because we want to get a meteorological data driven prediction model of initial flowering period. (Line 285)
- Lines 212 -213. ...the degree of dispersion is large (How much?)
Response: Thanks for the reviewer’s comment. We have added the value of dispersion to the manuscript. (Line 286-287)
- Line 215. Improve the writing of value/d in table 2 by value (d).
Response: Thanks for the reviewer’s comment. We have improved the writing here. (Line 289)
- Line 217. With an increase in the training epoch... Explain better what the training time means.
Response: Thanks for the reviewer’s comment. We have added the explanation of training time to the manuscript. (Line 292)
- Line 229. Figure 5 shows that... Isn't it Figure 4?
Response: Thanks for the reviewer’s comment. We have corrected this error, and again thank the reviewer for his suggestion. (Line 304)
- Lines 238-242 - How many sample factors were used? These samples refer to which data and locations? Write in the text.
Response: Thanks for the reviewer’s comment. We used the data of all samples including training dataset and test dataset, which is a matrix of 89 meteorological factors and ordinal number of initial flowering period. And we have added these to the manuscript. (Line 313-316)
- Line 243. Figure 5 (a), (b) and (c) reflect the contribution rate stability of each factor in different samples. Specify the number of samples used! Improve the explanation of Figure 5. It is confusing! What does each axis represent? What does the color palette mean? The results are good, but the explanation needs more work to show the importance of the technique. One way is to make the figures more detailed and explore the results more.
Response: Thanks for the reviewer’s comment. We have improved the sentence and added more explanation to the manuscript. (Line 313-316)
- Line 246. What are the values of the contribution rates of the models?
Response: Thanks for the reviewer’s comment. We have added more explanation to the manuscript. (Line 323-326)
- Line 250-251 .... the average factor contribution rate is different (?) How much is this difference?
... temperature factors are more important to the mode? Why? How did you get this result? From figure 5?
Response: Thanks for the reviewer’s comment. We have added the data of the difference of average factor contribution rate and explained why the temperature factors are more important. (Line 333-334)
- Line 254. ... GRU factors is greater than... Greater than how much?
Response: Thanks for the reviewer’s comment. We have added the data of absolute contribution rate of RNN, LSTM and GRU to the manuscript. (Line 337-340)
- Line 260. Make it clear what A1 to A37 are? x1-x89? writing that they are samples and weather factors does not justify it! What are these factors? What are these samples? From which cities?
Response: Thanks for the reviewer’s comment. We have adjusted the position of these the more detailed explanation, so we have deleted this sentence here. (Line 320-326)
- Line 262 - 265: The traditional prediction model of the flowering period is often selected for prediction by establishing a multiple regression prediction model... The sentence is very confusing! Rewrite it! What is SPSS? What degree of influence of various factors is this? What factors are these? Very badly written! Multiple linear regression? Explain this in the methodology!
Response: Thanks for the reviewer’s comment. We have rewritten the sentence to explant it better. (Line 346-348)
- Line 265 - However, there is an upper limit for the prediction accuracy of simple linear models [28,29]. What is the value of the upper limit for the prediction accuracy? Put it in the text!
Response: Thanks for the reviewer’s comment. Due to the mistakes in the references, it led to misunderstanding. This sentence means that the prediction effect of the linear model is lower than that of the nonlinear model, and it is associated with the next sentence. Therefore, we modified the sentence and reference to improve our writing. (Line 348-349)
- Line 266 - Chen and others have established linear and nonlinear models between the cherry flowering period and climate factors (Which ones?) and determined that they have a good simulation effect (What effect 'is that?) for the nonlinear mode (?). Very confusing sentence! Rewrite it!
Response: Thanks for the reviewer’s comment. We have added more explanation of linear and nonlinear modes, and we have also added the effect of nonlinear modes of cherry flowering period prediction. (Line 349-352)
- Line 268- 270: In the neural network structure of deep learning, there are linear operations (Which?) with a convolution layer as the core and nonlinear operations (Which?) with an activation function (?) as the core. Rewrite!
Response: Thanks for the reviewer’s comment. We have improved the sentence to avoid ambiguity. (Line 352-354)
- Line 273. According to the research of most scholars (Cite them)...we use the effective accumulated temperature of ≥0℃, ≥3℃, ≥5℃, and ≥10℃. Now that's what's hard to understand! What are these confusing variables? Rewrite!
Response: Thanks for the reviewer’s comment. We have rewritten it to avoid ambiguity. (Line 357)
- Line 276. x1 >= 0°C, x2 >=3°C... You accumulate this to infinity! I can't understand this! Rewrite.
Response: Thanks for the reviewer’s comment. We have rewritten it to avoid ambiguity. (Line 361-364)
- Line 278. According to the deep learning models and multiple linear regression model (Table 4)... Include the table in the text.
Response: Thanks for the reviewer’s comment. In the latest manuscript, we have added tables. (Line 367)
- Line 282. the accuracy of the deep learning model was significantly higher than... What is the confidence level to say it is significant? Put in the text!
Response: Thanks for the reviewer’s comment. We have added the confidence level to say it’s significant. (Line 370)
- Line 285. Due to the obvious spatial relationship of meteorological elements...With what is this spatial relationship obvious? Confusing! Write better! You use the word obvious a lot!
Response: Thanks for the reviewer’s comment. We have improved writing. For obvious relationships, we will provide data. (Line 375)
- 287. Cite Figure 6 in the text!
Response: Thanks for the reviewer’s comment. We have cited the Figure 6 to the explanation of it. (Line 380)
- Lines 288-289. the prediction results of the RNN model lag behind the true value, mainly focusing on [-2d, -1d) and [-3d, -2d), while the prediction error of LSTM and GRU are mainly focused on [-1d, 0d), but the error of LSTM results exceeds 3d. Confusing writing! What is the true value? The date of bloom observation? What does the interval [-2d, -1d) mean? Explain in the text! What does -2d mean? Underestimated by 2 days? 0d is an error or a model hit?
Response: Thanks for the reviewer’s comment. We have improved the sentence and added more explanations. (Line 377-379)
- Line 297. the three models show similar characteristics... What are these similar characteristics? Write in the text!
Response: Thanks for the reviewer’s comment. Similar characteristics have been added after this sentence, and we have modified this sentence to avoid ambiguity. (Line 389-390)
- Line 298 - Julian days gradually increase from low latitudes to high latitudes. Would not be days of flowering? Julian day is equal to days occurring in a year. So, improve it there! Low latitudes to high latitudes (What are the low and high latitude values?) Write in the text. ...obvious hierarchical structure? What is this obvious hierarchical structure? What do you mean!
Response: Thanks for the reviewer’s comment. We have added more explanations and improved the writing to help readers better understand the meaning of this paragraph. (Line 390-393)
- Lines 299-301 I don't know China, so please identify in Figure 7 Mongolia and the Yangtze River Delta? Also add on all maps the latitude and longitude and the direction from North!
Response: Thanks for the reviewer’s comment. We have modified Figure 1 and added more explanations to manuscript. (Line 129)
- Line 305 - 307 What is the type of humidity? What is the type of precipitation? Why are the distributions of air temperature and humidity uneven? What are the overarching factors that affect the flowering period? Explain in the text.
Response: Thanks for the reviewer’s comment. The sentence here expressed the possibility of affecting the distribution of initial flowering period in China, but we cannot provide scientific proof, so we have deleted this sentence.
- Line 314 -315. flowering prediction models... Give examples of the traditional flowering prediction model. These traditional flowering prediction models were not well explained throughout the text. Improve it!
Response: Thanks for the reviewer’s comment. We have improved the sentence and added the explanation to the manuscript. (Line 411)
- Line 318-320. Therefore, the mathematical regression models are built around accumulated temperature, average temperature and other factors related to temperature. Specify if the temperature is of the canopy surface air! Name these other factors!
Response: Thanks for the reviewer’s comment. We have added the definition of the temperature here. (Line 415-417)
- ... models may have obvious error ? Cite the obvious errors!
Response: Thanks for the reviewer’s comment. We have modified the sentence here. (Line 417-418)
- Line 321. ... the traditional prediction model cannot realize the nationwide prediction of initial flowering. Explain why?
Response: Thanks for the reviewer’s comment. We have deleted the sentence in order to avoid ambiguity, which means that the prediction effect of traditional model in large region in less than DL models according to MAE, MAPE and R2. This is the result of data, and we didn’t delve into the interpretation of this result.
- Line 326 - ... a better effect in the prediction of multiple species. What do you mean by this? Confusing! The model can only be applied to multiple species and not to a single species?
Response: Thanks for the reviewer’s comment. We have deleted the sentence because it does not come from our conclusions, but from our expectations of the DL models.
- Line 328. ... the atmospheric environment ?
Response: Thanks for the reviewer’s comment. We believe that carbon emission is also one of the factors affecting flowering, and carbon emission in an important research element of environmental science. However, this statement is ambiguous, so we have deleted the term of atmospheric environment. (Line 423-424)
Thanks again for your suggestions.
look forward to working with you to move the manuscript closer to publication in the Agriculture.
Sincerely yours
Round 2
Reviewer 1 Report
Review of the submitted manuscript entitled Research on a Prediction Model of the Initial Flowering Period of Arborvitae Based on Deep Learning Algorithms
The authors have improved the manuscript. The research, I believe, is technically well justified in terms of modeling, and the Discussion and Conclusions need minor revisions. However, I have comments on the meteorological observations themselves. In addition, the manuscript still needs many corrections, mainly technical. I am not a native speaker, but I feel that a thorough language correction is needed for the entire text. The convoluted and incorrect sentence construction and inappropriate or inadequate terms make the article difficult to read. I only added some suggestions because taking care of the language's quality is the author's, not the reviewer's responsibility.
The title is long and awkward. The word "Research on" seems unnecessary because scientific journals usually publish research about something. The authors should consider correcting it, such as "Utility of Deep Learning Algorithms in the Phenological Prediction Models".
Use of species names: a good standard widely accepted is to give the full name and the author abbreviation when it is first used in the text, e.g. Platycladus orientalis (L.) Franco. After that, the abbreviated name P. orientalis should be used. This rule separately applies to the abstract and the main text.
Keywords should not include the words used in the title. Their function is to carry additional information about the article's content that is not included in the title. Their better selection allows for better positioning of the article in web browsers and can impact the citability of the publication.
Introduction:
L.38: This claim is valid for plants with short flowering times. Note that many species bloom from early spring to late fall. Some flowering plants can be found under the snow.
L.39: "ecological health" sounds very strange. What would it mean? The authors should replace the controversial term with another one.
L.42-44: This sentence does not contribute any necessary information. I suggest removing it.
L.45: "According to knowledge of phenological ecology, climatic conditions have ..." -> "The climatic conditions have ..."
L.46: Also, soil temperature is relevant to phenology.
L.48: "traditional statistical equations" sounds strange. I suggest removing "traditional."
L.72: Which disasters? Add information about this. This could be important.
L.80: "moisture content" -> "moisture" By the way, soil moisture is an important factor affecting spring phenology. Water deficiency can delay it as many studies have shown. The authors should mention this.
L.81-84: This sentence is written in a poorly understood way.
L.84": "... conducted applicability ..." -> "... conducted ..."
L.86-87: "organic phenological model" may not be understood by the reader. It sounds strange.
L.96: The authors can add „A solution for spatial phenology modelling may be to model phenology using herbarium and Citizen Science records and gridded climatic data. Recently, the flowering of Anemone nemorosa was modelled in this way across Europe. However, this approach has some limitations related to the availability of replicated phenological observations and Spatial and taxonomic biases (Puchałka et al. 2022). Hence, long-term local monitoring data are still invaluable in phenological studies.
Puchałka, R., Klisz, M., Koniakin, S., Czortek, P., Dylewski, Ł., Paź-Dyderska, S., Vitkova, M., Sadlo, J., Rasomavicius, V., Carni, A., De Sanctis, M., Dyderski, M., 2022. Citizen science helps predictions of climate change impact on flowering phenology: A study on Anemone nemorosa. Agric. For. Meteorol. 325, 109133. https://doi.org/10.1016/j.agrformet.2022.109133
L.96-99: This is a long and difficult-to-read sentence. It would be worth simplifying the message, e.g. "In our research, we demonstrate the capabilities of deep learning algorithms that have so far been used to a limited extent in phenological research. We believe that the results obtained in our study will find wide application and contributing to a better understanding of the phenological response of plants to meteorological conditions.
L.108: "2.1 Research object" -> "2.1 Studied species"
L.109: Unnecessary subsection.
L.110: This paragraph can be started with "Platycladus orientalis (L.) Franco (Cupressaceae)" ..., or with P". orientalis (Cupressaceae) ..." if the full name is used before.
L.110-119: This is a very chaotic and too general description of the species. The characteristics of this species, in what climate it occurs, and how it is distributed should be specified more precisely. In addition, there need to be more literature citations for the information given here. One can cite Flora of China, where the authors can find enough descriptions of the species and its distribution.
L.151: The figures are unsightly and illegible. All names of administrative units and climatic regions should be removed from the figures and included in the figure caption.
L.157: GYMNOSPERMS DO NOT HAVE PETALS! I wonder if the authors got it wrong or are unfamiliar with plant morphology. This calls into question the credibility of this research. The methodology for identifying phenological phases should be explained in detail in Materials and Methods.
L.480: It makes no sense to caption the chart "Chart". The reader should note that the figure represents a chart.
L:555: „Traditional flowering perdition …” There is no such thing. There are many methodological approaches, and I doubt if you can talk about tradition here.
L.579-580: "collinearity diagnosis" sounds strange.
L.678-680: Interpretation of results should be moved from Results to Discussion.
L.695: How does prediction relate to inter-annual variability in meteorological conditions? What about climate change?
L.718-719: Explain why these factors can cause flowers to open. Is this a mental shortcut? Cite published studies.
L.752-754: Too trivial to include such information in Conclusions.
L.762: "meteorological disasters" Unclear what the authors mean.
Kind regards
Author Response
Thank the reviewer for spending valuable time reviewing the manuscript for us and putting forward useful comments. We have revised the manuscript with comments, hoping to meet the requirements of Agriculture. Thanks again.
- 1. The convoluted and incorrect sentence construction and inappropriate or inadequate terms make the article difficult to read.
Response: Thanks for the reviewer’s comment. We have modified or added more explanations to some difficult sentences in the article
- The title is long and awkward. The word "Research on" seems unnecessary because scientific journals usually publish research about something. The authors should consider correcting it, such as "Utility of Deep Learning Algorithms in the Phenological Prediction Models".
Response: Thanks for the reviewer’s comment. We agree with you and have modified the title. (Line 2-3)
- Use of species names: a good standard widely accepted is to give the full name and the author abbreviation when it is first used in the text, e.g. Platycladus orientalis (L.) Franco. After that, the abbreviated name P. orientalis should be used. This rule separately applies to the abstract and the main text.
Response: Thanks for the reviewer’s comment. We have modified the manuscript according to your suggestion. (Line 13)
- Keywords should not include the words used in the title. Their function is to carry additional information about the article's content that is not included in the title. Their better selection allows for better positioning of the article in web browsers and can impact the citability of the publication.
Response: Thanks for the reviewer’s comment. We have modified the keyword according to the comment. (Line 33-34)
- L.38: This claim is valid for plants with short flowering times. Note that many species bloom from early spring to late fall. Some flowering plants can be found under the snow.
Response: Thanks for the reviewer’s comment. To avoid ambiguity, we have emphasized that the research work is mainly focused on plants with short flowering times. (Line 39-40)
- L.39: "ecological health" sounds very strange. What would it mean? The authors should replace the controversial term with another one.
Response: Thanks for the reviewer’s comment. Here we want to describe the ecological damage caused by climate change. For a clearer explanation, we have modified it. (Line 38)
- L.42-44: This sentence does not contribute any necessary information. I suggest removing it.
Response: Thanks for the reviewer’s comment. We have deleted this sentence.
- L.45: "According to knowledge of phenological ecology, climatic conditions have ..." -> "The climatic conditions have ..."
Response: Thanks for the reviewer’s comment. We agree with you and have modified this sentence. (Line 43)
- L.46: Also, soil temperature is relevant to phenology.
Response: Thanks for the reviewer’s comment. We have added soil temperature to emphasize its importance. (Line 44)
- L.48: "traditional statistical equations" sounds strange. I suggest removing "traditional."
Response: Thanks for the reviewer’s comment. We have deleted the “traditional”. (Line 46)
- L.72: Which disasters? Add information about this. This could be important.
Response: Thanks for the reviewer’s comment. We have added more explanations to the manuscript. (Line 57-58)
- L.80: "moisture content" -> "moisture" By the way, soil moisture is an important factor affecting spring phenology. Water deficiency can delay it as many studies have shown. The authors should mention this.
Response: Thanks for the reviewer’s comment. We have modified the sentence, and added more explanations to the manuscript. (Line 65-67)
- L.81-84: This sentence is written in a poorly understood way.
Response: Thanks for the reviewer’s comment. We have modified the sentence to make it more clear. (Line 67-68)
- L.84": "... conducted applicability ..." -> "... conducted ..."
Response: Thanks for the reviewer’s comment. We have modified this sentence. (Line 69)
- L.86-87: "organic phenological model" may not be understood by the reader. It sounds strange.
Response: Thanks for the reviewer’s comment. We have modified this word and added more explanation about the model. (Line 71-72)
- L.96: The authors can add „A solution for spatial phenology modelling may be to model phenology using herbarium and Citizen Science records and gridded climatic data. Recently, the flowering of Anemone nemorosa was modelled in this way across Europe. However, this approach has some limitations related to the availability of replicated phenological observations and Spatial and taxonomic biases (Puchałka et al. 2022). Hence, long-term local monitoring data are still invaluable in phenological studies.
Puchałka, R., Klisz, M., Koniakin, S., Czortek, P., Dylewski, Ł., Paź-Dyderska, S., Vitkova, M., Sadlo, J., Rasomavicius, V., Carni, A., De Sanctis, M., Dyderski, M., 2022. Citizen science helps predictions of climate change impact on flowering phenology: A study on Anemone nemorosa. Agric. For. Meteorol. 325, 109133. https://doi.org/10.1016/j.agrformet.2022.109133
Response: Thanks for the reviewer’s comment. We agree with you very much and have modified the manuscript according to the suggestions. (Line 78-83)
- L.96-99: This is a long and difficult-to-read sentence. It would be worth simplifying the message, e.g. "In our research, we demonstrate the capabilities of deep learning algorithms that have so far been used to a limited extent in phenological research. We believe that the results obtained in our study will find wide application and contributing to a better understanding of the phenological response of plants to meteorological conditions.
Response: Thanks for the reviewer’s comment. We quite agree with your suggestion and have modified our manuscript. (Line 87-90)
- L.108: "2.1 Research object" -> "2.1 Studied species" and L.109: Unnecessary subsection.
Response: Thanks for the reviewer’s comment. We have modified the manuscript. Since readers may not know about China, we keep Region here. (Line 95 and Line 106)
- L.110: This paragraph can be started with "Platycladus orientalis (L.) Franco (Cupressaceae)" ..., or with P". orientalis (Cupressaceae) ..." if the full name is used before.
Response: Thanks for the reviewer’s comment. We have modified it to the manuscript. (Line 96)
- L.110-119: This is a very chaotic and too general description of the species. The characteristics of this species, in what climate it occurs, and how it is distributed should be specified more precisely. In addition, there need to be more literature citations for the information given here. One can cite Flora of China, where the authors can find enough descriptions of the species and its distribution.
Response: Thanks for the reviewer’s comment. We have modified it according to Flora of China. (Line 96-105)
- L.151: The figures are unsightly and illegible. All names of administrative units and climatic regions should be removed from the figures and included in the figure caption.
Response: Thanks for the reviewer’s comment. We have modified Figure 1. (Line 127-135)
- L.157: GYMNOSPERMS DO NOT HAVE PETALS! I wonder if the authors got it wrong or are unfamiliar with plant morphology. This calls into question the credibility of this research. The methodology for identifying phenological phases should be explained in detail in Materials and Methods.
Response: Thanks for the reviewer’s comment. The definition of the initial flowering period we used is a broad one. For Platycladus orientalis, there is a sporophyll ball, which has no petals, stamens and other structures, and will still be considered as "false flowers". Therefore, during phenological observation, the flowering data of Platycladus orientalis will be collected. To avoid ambiguity, we have modified the manuscript. (Line 138-139)
- L.480: It makes no sense to caption the chart "Chart". The reader should note that the figure represents a chart.
Response: Thanks for the reviewer’s comment. We have modified the non-standard title in the manuscript. (Line 278, Line 305, Line 317 and Line 349-351)
- L:555: „Traditional flowering perdition …” There is no such thing. There are many methodological approaches, and I doubt if you can talk about tradition here.
Response: Thanks for the reviewer’s comment. What we want to compare here is the prediction difference between DL model and non-DL model. We agree with your suggestion, so we have modified this sentence. (Line 353)
- L.579-580: "collinearity diagnosis" sounds strange.
Response: Thanks for the reviewer’s comment. We have analyzed the collinearity of different variables here. To avoid ambiguity, we have modified this sentence. (Line 377-378)
- L.678-680: Interpretation of results should be moved from Results to Discussion.
Response: Thanks for the reviewer’s comment. We have revised the manuscript, thank you again for your comment. (Line 429-432)
- L.695: How does prediction relate to inter-annual variability in meteorological conditions? What about climate change?
Response: Thanks for the reviewer’s comment. We have added the interannual change of florescence to the manuscript, and added the explanation of the temporal and spatial differences of climate change in Discussion. (Line 412-413, Line 431-432)
- L.718-719: Explain why these factors can cause flowers to open. Is this a mental shortcut? Cite published studies.
Response: Thanks for the reviewer’s comment. We have added more explanation about why temperature can cause flowers by citing published studies. (Line 423-424)
- L.752-754: Too trivial to include such information in Conclusions.
Response: Thanks for the reviewer’s comment. We have modified the Conclusions to summed up more important results. (Line 449-460)
- L.762: "meteorological disasters" Unclear what the authors mean.
Response: Thanks for the reviewer’s comment. We have modified the sentence to avoid unclear explanations. (Line 462-463)
Sincerely yours
Reviewer 2 Report
I accept the article after corrections.
Author Response
Thank you very much for the approval. You have provided a lot of intentional review comments during the review process. We hope this manuscript can meet the requirements of Agriculture。